# A Geometric Structure of Acceleration and Its Role in Making Gradients Small Fast

**Jongmin Lee**
Seoul National University
dlwhd2000@snu.ac.kr

**Chanwoo Park**
Seoul National University
chanwoo.park@snu.ac.kr

**Ernest K. Ryu**
Seoul National University
ernestryu@snu.ac.kr

## Abstract

Since Nesterov's seminal 1983 work, many accelerated first-order optimization methods have been proposed, but their analyses lacks a common unifying structure. In this work, we identify a geometric structure satisfied by a wide range of first-order accelerated methods. Using this geometric insight, we present several novel generalizations of accelerated methods. Most interesting among them is a method that reduces the squared gradient norm with $\mathcal{O}(1/K^4)$ rate in the prox-grad setup, faster than the $\mathcal{O}(1/K^3)$ rates of Nesterov's FGM or Kim and Fessler's FPGM-m.

## 1 Introduction

Since Nesterov's seminal 1983 work [57], accelerated methods in optimization have been widely used in large-scale optimization and machine learning. However, the many accelerated methods have been developed and analyzed with disparate techniques, without a unified framework.

In this work, we identify geometric structures, which we call the parallel and collinear structures, satisfied by a wide range of gradient- and prox-based accelerated methods including: Nesterov's FGM [57], OGM [47], OGM-G [47], Nesterov's FGM in the strongly convex setup (SC-FGM) [60, (2.2.22)], SC-OGM [63], TMM [78], non-stationary SC-FGM [19, §4.5], ITEM [73], geometric descent [13], Güler's first and second accelerated proximal methods [37], and FISTA [11] Using the insight provided by these geometric structures, we present several novel generalizations of accelerated methods.

Among these novel generalizations, the most interesting is FISTA-G, which reduces the squared gradient norm with $\mathcal{O}(1/K^4)$ rate when combined with FISTA in the prox-grad (composite minimization) setup. The rate is optimal as it matches the $\Omega(1/K^4)$ complexity lower bound [55, 56] and is faster than the $\mathcal{O}(1/K^3)$ rates achieved by of Nesterov's AGM [46, 69] in the smooth convex setup or Kim and Fessler's FPGM-m [45] in the prox-grad setup.

### 1.1 Preliminaries and notations

For $L > 0$, $f \colon \mathbb{R}^n \to \mathbb{R}$ is $L$-smooth if $f$ is differentiable and
$$\|\nabla f(x) - \nabla f(y)\| \leq L\|x - y\| \qquad \forall\, x, y \in \mathbb{R}^n.$$
For $\mu > 0$, $g \colon \mathbb{R}^n \to \mathbb{R} \cup \{\infty\}$ is $\mu$-strongly convex if $g(x) - (\mu/2)\|x\|^2$ is convex. Let $L$ and $\mu$ respectively denote the smoothness and strong convexity parameters of $f$. Write the gradient steps with stepsizes $1/L$ and $1/\mu$ as
$$x^+ = x - \frac{1}{L}\nabla f(x), \qquad x^{++} = x - \frac{1}{\mu}\nabla f(x).$$
For $\lambda > 0$, the proximal operator [53, 62] $\mathrm{Prox}_{\lambda g} \colon \mathbb{R}^n \to \mathbb{R}^n$ is defined as
$$x^\circ = \mathrm{Prox}_{\lambda g}(x) = \operatorname*{arg\,min}_{y \in \mathbb{R}^n} \left\{ g(y) + \frac{1}{2\lambda}\|y - x\|^2 \right\}.$$

35th Conference on Neural Information Processing Systems (NeurIPS 2021).

When $g$ is closed, convex, and proper [65], the proximal operator is well-defined i.e., the $\arg\min$ exists and is unique [52]. Define the prox-grad step as

$$x^\oplus = \underset{y \in \mathbb{R}^n}{\arg\min} \left\{ f(x) + \langle \nabla f(x), y - x \rangle + g(y) + \frac{L}{2} \|y - x\|^2 \right\}.$$

If $g = 0$, then $x^\oplus = x^+$. If $f = 0$ and $1/\lambda = L$, then $x^\oplus = x^\circ$. Furthermore, write

$$\tilde{\nabla}_L F(x) := -L(x^\oplus - x), \qquad \tilde{\nabla}_{1/\lambda} g(x) := -\frac{1}{\lambda}(x^\circ - x).$$

$u \in \mathbb{R}^n$ is a subgradient of convex function $g$ at $x$ if

$$g(x) + \langle u, y - x \rangle \leq g(y) \qquad \forall\, y \in \mathbb{R}^n.$$

Subdifferential of $g$ at $x$, denoted $\partial g(x)$, is the set of subgradients of $g$ at $x$. Throughout this paper, we consider the problem

$$\underset{x \in \mathbb{R}^n}{\text{minimize}} \quad F(x) := f(x) + g(x), \tag{P}$$

where $f \colon \mathbb{R}^n \to \mathbb{R}$ is convex and $L$-smooth and $g \colon \mathbb{R}^n \to \mathbb{R} \cup \{\infty\}$ is closed, convex, and proper. We say we are in the smooth convex setup if $f \neq 0$ and $g = 0$, the proximal-point setup if $f = 0$ and $g \neq 0$, and the prox-grad setup if $f \neq 0$ and $g \neq 0$. Write $f_\star$, $g_\star$, and $F_\star$ to respectively denote the infima of $f$, $g$, and $F$. We also informally assume $\text{Prox}_{\lambda g}$ is efficient to evaluate [17].

## 1.2  Prior works

In convex optimization and machine learning, the classical goal of algorithms is to reduce the function value efficiently. In the smooth convex setup, Nesterov's celebrated fast gradient method (FGM) [57] achieves an accelerated rate, and the optimized gradient method (OGM) [44] improves this rate by a factor of 2, which is in fact exactly optimal [27]. In the smooth strongly convex setup, the strongly convex fast gradient method (SC-FGM) [60, (2.2.22)], strongly convex optimized gradient method (SC-OGM) [63], and non-stationary SC-FGM [19, §4.5] achieves an accelerated rate, and the triple momentum method (TMM) [78] and information theoretic exact method (ITEM) [73] achieve exact optimal rates [28]. In the proximal-point setup, there are Güler's first and second accelerated proximal methods [37]. There are also other variants of accelerated proximal methods [51, 49, 39, 77].

The study of algorithms for reducing the gradient magnitude, which can help us understand non-convex optimization better and design faster non-convex machine learning algorithms, was initiated by Nesterov [58]. For smooth nonconvex minimization, gradient descent (GD) achieves $\mathcal{O}((f(x_0) - f_\star)/K)$ rate [54, Proposition 3.3.1]. In the smooth convex setup, combining $K$ iterations of FGM with $K$ iterations of GD achieves $\mathcal{O}(\|x_0 - x_\star\|^2/K^3)$ rate [58]. Adding a strongly convex regularization and then using SC-FGM achieves $\tilde{\mathcal{O}}(\|x_0 - x_\star\|^2/K^4)$ rate, where $\tilde{\mathcal{O}}$ ignores logarithmic factors [58]. Finally, OGM-G [47] achieved $\mathcal{O}((f(x_0) - f_\star)/K^2)$ rate and $\mathcal{O}(\|x_0 - x_\star\|^2/K^4)$ rate is achieved by combining FGM with OGM-G [61, Remark 2.1]. This rate is optimal as it matches the $\Omega(\|x_0 - x_\star\|^2/K^4)$ lower bound [55, 56]. Making gradients small have also been studied in the setup with stochastic gradients [2, 3, 33, 79, 34], for composite problems with strong convexity and non-Euclidean norms [23], and for convex-concave saddle-point problems [22, 24, 80].

In the prox-grad setup, iterative shrinkage-thresholding algorithm (ISTA) [12, 64, 20, 18] achieves $\mathcal{O}(\|x_0 - x_\star\|^2/K)$ rate on function-value suboptimality and fast iterative shrinkage-thresholding algorithm (FISTA) [11] accelerates this rate to $\mathcal{O}(\|x_0 - x_\star\|^2/K^2)$. On the squared gradient magnitude, FPGM-m achieves $\mathcal{O}(\|x_0 - x_\star\|^2/K^3)$ rate [45].

The performance estimation problem (PEP) is a computer-assisted proof methodology based on semidefinite programming [29, 75, 74]. Extensions and variations of the PEP have been utilized to obtain analyses and algorithms that would be difficult to obtain without the assistance of computers [67, 72, 10, 76, 21, 36, 50, 26, 43]. In particular, OGM [29, 44], OGM-G [47], and ITEM [73] were obtained with the PEP. Integral quadratic constraints is a related computer-assisted approach based on control-theoretic ideas [48, 38, 78, 31].

A Lyapunov analysis constructs a nonincreasing quantity, and many modern analyses of accelerated first-order methods are based on this technique [11, 70, 9, 72, 1, 6, 7, 8, 63]. In fact, Nesterov's original presentation of FGM was a Lyapunov analysis [57].

Finally, we mention some closely related prior work. The scaled relative graph (SRG) analyzes optimization algorithms via Euclidean geometry [66, 40, 41, 68], but the SRG has not been used to analyze accelerated methods. Linear coupling [4, 5] interprets acceleration as a coupling between gradient descent and mirror descent to efficiently reduce function values. Geometric descent [13, 16, 42] is an accelerated algorithm designed expressly based on geometric principles, and quadratic averaging [30] is an equivalent algorithm with an alternate interpretation. The method of similar triangles (MST) generalizes FGM with geometric notions [32, 60, 1]. The parallel and collinear structures of this work, in our view, expand these prior notions and concretely articulate ideas that had been utilized implicitly. In particular, the prior interpretations, as is, are insufficient for analyzing OGM-G and deriving our newly proposed method FISTA-G. In Section H of the appendix, we further discuss the relationship of our contributions with these prior works.

### 1.3 Contribution

This paper presents two major contributions, one conceptual and one concrete. The first is the identification of the parallel and collinear geometric structures, which are observed in a wide variety of accelerated first-order methods. This geometric structure provides us with valuable insight into the mechanism of acceleration and enables us to obtain results that would be otherwise difficult to discover, including FISTA-G. The second major contribution is the novel method FISTA-G and FISTA+FISTA-G. To the best of our knowledge, FISTA+FISTA-G is the first method to achieve the rate $\mathcal{O}(1/K^4)$ on the squared gradient norm in the prox-grad setup.

In addition, we use the geometric structures to find G-FISTA-G, G-FGM-G, G-Güler-G, Proximal-TMM, and Proximal-ITEM, novel variants of accelerated first-order methods, and present them as minor contributions. (G-FGM-G is the abbreviation for Generalized-FGM-(for reducing Gradients).)

## 2 Making gradients small at rate $\mathcal{O}(1/K^4)$ in the prox-grad setup

In this section, we present a novel method FISTA-G, which reduces the squared gradient norm with $\mathcal{O}(1/K^4)$ rate when combined with FISTA in the prox-grad setup. Our analysis uses a novel and unusual Lyapunov function, whose discovery crucially relied on the geometric insights presented later in Section 3. Here, we provide a self-contained analysis that uses, but does not explain this Lyapunov function.

**Smooth convex setup.** Kim and Fessler's (**OGM-G**) [47] is

$$x_{k+1} = x_k^+ + \frac{(\theta_k - 1)(2\theta_{k+1} - 1)}{\theta_k(2\theta_k - 1)}(x_k^+ - x_{k-1}^+) + \frac{2\theta_{k+1} - 1}{2\theta_k - 1}(x_k^+ - x_k) \quad \text{for } k = 0, 1, \ldots, K-1,$$

where $x_{-1}^+ := x_0$, $\theta_K = 1$, $\theta_k = \frac{1+\sqrt{1+4\theta_{k+1}^2}}{2}$ for $k = 1, 2, \ldots, K-1$, and $\theta_0 = \frac{1+\sqrt{1+8\theta_1^2}}{2}$. Note, $K$ is the total number of iterations and $\theta_0 = \mathcal{O}(K)$, so $\theta_0$ is a function of $K$. OGM-G is the first method to achieve the accelerated rate $\mathcal{O}((f(x_0) - f_\star)/K^2)$ on the squared gradient norm, and the method was originally obtained through a computer-assisted methodology. However, the computer-generated analysis is verifiable but arguably difficult to understand.

We characterize the convergence of OGM-G with the following novel Lyapunov function

$$U_k = \frac{1}{\theta_k^2}\left(\frac{1}{2L}\|\nabla f(x_K)\|^2 + \frac{1}{2L}\|\nabla f(x_k)\|^2 + f(x_k) - f(x_K) - \langle\nabla f(x_k), x_k - x_{k-1}^+\rangle\right)$$

$$+ \frac{L}{\theta_k^4}\langle z_k - x_{k-1}^+, z_k - x_K^+\rangle, \quad \text{for } k = 1, 2, \ldots, K,$$

where $z_k = x_k + \frac{(\theta_k - 1)^2}{2\theta_k - 1}(x_k - x_{k-1}^+)$ for $k = 1, 2, \ldots, K$, (note $z_K = x_K$) through the steps

$$\frac{1}{L}\|\nabla f(x_K)\|^2 = U_K \le U_{K-1} \le \cdots \le U_1 \le \frac{2}{\theta_0^2}\left(\frac{1}{2L}\|\nabla f(x_K)\|^2 + f(x_0) - f(x_K)\right).$$

Section 3 explains the geometric insights behind the discovery of this Lyapunov function and the proof that $\{U_k\}_{k=1}^K$ is nonincreasing. As related work, Diakonikolas and Wang [24] presented a human-understandable analysis of OGM-G, but their analysis is not a Lyapunov analysis since (as they acknowledge) they do not establish a nonincreasing quantity. In any case, our main contribution of this section is the following generalization of the method and analysis to the prox-grad setup.

## 2.1 Accelerated $\mathcal{O}((F(x_0) - F_\star)/K^2)$ rate with FISTA-G

We now present the novel method (**FISTA-G**):

$$x_{k+1} = x_k^\oplus + \frac{\varphi_{k+1} - \varphi_{k+2}}{\varphi_k - \varphi_{k+1}}(x_k^\oplus - x_{k-1}^\oplus) \qquad \text{for } k = 0, 1, \ldots, K-1,$$

where $x_{-1}^\oplus := x_0$, $\varphi_{K+1} = 0$, $\varphi_K = 1$, and

$$\varphi_k = \frac{\varphi_{k+2}^2 - \varphi_{k+1}\varphi_{k+2} + 2\varphi_{k+1}^2 + (\varphi_{k+1} - \varphi_{k+2})\sqrt{\varphi_{k+2}^2 + 3\varphi_{k+1}^2}}{\varphi_{k+1} + \varphi_{k+2}} \qquad \text{for } k = -1, 0, \ldots, K-1.$$

Note that $K$ is the total number of iterations.

**Theorem 1.** *Consider* (P). *FISTA-G's final iterate $x_K$ exhibits the rate*

$$\min \left\| \partial F(x_K^\oplus) \right\|^2 \le 4 \left\| \tilde{\nabla}_L F(x_K) \right\|^2 \le \frac{264L}{(K+2)^2}(F(x_0) - F_\star).$$

We clarify that $\min \|\partial F(x)\|^2 = \min\{\|u\|^2 \mid u \in \partial F(x)\}$ for $x \in \mathbb{R}^n$.

*Proof outline.* Define $z_0 = x_0$, $z_k = \frac{\varphi_k}{\varphi_k - \varphi_{k+1}}x_k - \frac{\varphi_{k+1}}{\varphi_k - \varphi_{k+1}}x_{k-1}^\oplus$ for $k = 0, 1, \ldots, K$, and

$$U_k = \frac{2\varphi_{k-1}}{(\varphi_{k-1} - \varphi_k)^2}\left(\frac{1}{2L}\left\|\tilde{\nabla}_L F(x_k)\right\|^2 + F(x_k^\oplus) - F(x_K^\oplus) - \left\langle \tilde{\nabla}_L F(x_k), x_k - x_{k-1}^\oplus \right\rangle \right)$$

$$+ \frac{L}{\varphi_k}\left\langle z_k - x_{k-1}^\oplus, z_k - x_K^\oplus \right\rangle \qquad \text{for } k = 0, 1, \ldots, K.$$

(Note that $z_K = x_K$.) We can show that $\{U_k\}_{k=0}^K$ is nonincreasing. Then

$$\frac{1}{2L}\left\|\tilde{\nabla}_L F(x_K)\right\|^2 = U_K \le \cdots \le U_0 = \frac{2\varphi_{-1}}{(\varphi_{-1} - \varphi_0)^2}\left(\frac{1}{2L}\left\|\tilde{\nabla}_L F(x_0)\right\|^2 + F(x_0^\oplus) - F(x_K^\oplus)\right)$$

$$\le \frac{2\varphi_{-1}}{(\varphi_{-1} - \varphi_0)^2}\left(F(x_0) - F(x_K^\oplus)\right) \le \frac{33L}{(K+2)^2}(F(x_0) - F_\star),$$

where we used $\frac{1}{2L}\left\|\tilde{\nabla}_L F(x_0)\right\|^2 \le F(x_0) - F(x_0^\oplus)$, stated as Lemma 9 in the appendix. $\qquad\square$

The complete proof is presented in Section E of the appendix. In fact, FISTA-G is the best instance within the family of methods G-FISTA-G presented in Section E of the appendix, in the sense that FISTA-G is the instance for which we could obtain the smallest constant in the bound.

## 2.2 Accelerated $\mathcal{O}(\|x_0 - x_\star\|^2/K^4)$ rate with FISTA+FISTA-G

Define the method (**FISTA+FISTA-G**) as: from a starting point $x_0$ run $K$ iterations of FISTA [11] and then from the output of FISTA start FISTA-G and run $K$ iterations. ($2K$ iterations total.) We denote the final iterate of this method as $x_{2K}$.

**Corollary 1.** *Consider* (P). *Assume $F$ has a minimizer $x_\star$. FISTA+FISTA-G's final iterate $x_{2K}$ exhibits the rate*[1]

$$\min \left\| \partial F(x_{2K}^\oplus) \right\|^2 \le 4 \left\| \tilde{\nabla}_L F(x_{2K}) \right\|^2 \le \frac{528L^2}{(K+2)^4}\|x_0 - x_\star\|^2.$$

*Proof.* Combining the $F(x_K^\oplus) - F_\star \le 2L\|x_0 - x_\star\|^2/(K+2)^2$ rate of FISTA [11, Theorem 4.4] with Theorem 1, we get the rate of FISTA+FISTA-G:

$$\min \left\| \partial F(x_{2K}^\oplus) \right\|^2 \le 4 \left\| \tilde{\nabla}_L F(x_{2K}) \right\|^2 \le \frac{264L}{(K+2)^2}(F(x_K^\oplus) - F_\star) \le \frac{528L^2}{(K+2)^4}\|x_0 - x_\star\|^2.$$

$$\square$$

---

[1] Obtaining an iterate $x_K$ with $\|\tilde{\nabla}_L F(x_K)\|^2 \le \epsilon$ requires $K \ge \left(\frac{66L(F(x_0) - F_\star)}{\epsilon}\right)^{\frac{1}{2}}$ iterations for FISTA-G, and $K \ge 2\left(\frac{132L^2\|x_0 - x_\star\|^2}{\epsilon}\right)^{\frac{1}{4}}$ iterations for FISTA+FISTA-G, where $K$ is a positive even integer.

FISTA+FISTA-G was inspired by the FGM+OGM-G method, which has $\mathcal{O}(\|x_0 - x_\star\|^2/K^4)$ rate on the squared gradient norm in the smooth convex setup [61, Remark 2.1]. Nesterov FGM by itself only achieves $\mathcal{O}(\|x_0 - x_\star\|^2/K^3)$ rate [75, 71, 46, 69, 24]. In the prox-grad setup, FPGM-m [45] held the prior state-of-the-art rate of $\mathcal{O}(\|x_0 - x_\star\|^2/K^3)$. Our $\mathcal{O}(\|x_0 - x_\star\|^2/K^4)$ rate matches the known complexity lower bound [55, 56] and therefore is optimal in the case $g = 0$.

## 3 Parallel structure for the convex setup

In this section, we present the *parallel structure*, a geometric structure observed in a wide range of accelerated first-order methods. We then utilize the parallel structure to obtain novel variants of accelerated first-order methods and, in particular, obtain the Lyapunov analysis presented in Section 2.

### 3.1 Parallel structure of acceleration

Nesterov's (**FGM**) [57] has the form:

$$x_{k+1} = x_k^+ + \frac{\theta_k - 1}{\theta_{k+1}}(x_k^+ - x_{k-1}^+)$$
$$z_k = x_k + (\theta_k - 1)(x_k - x_{k-1}^+) \qquad \text{for } k = 0, 1, \ldots,$$

where $x_{-1}^+ := x_0$, $x_0 = z_0$, $\theta_0 = 1$, and $\theta_{k+1} = \frac{1+\sqrt{1+4\theta_k^2}}{2}$ for $k = 0, 1, \ldots$. The $z_k$-iterates are known as the *auxiliary sequence*, and they play a key role in the Lyapunov analysis of FGM [57]. Figure 1 (left) depicts $x_{k-1}^+$, $x_k$, $x_k^+$, $x_{k+1}$, $z_k$, and $z_{k+1}$. These points in $\mathbb{R}^n$ lie on a 2D-plane, which we call *the plane of iteration*. Observe that the line segments representing $x_k^+ - x_k$ and $z_{k+1} - z_k$ are parallel. (We prove observations 1 and 2 in Section B of the appendix.)

**Observation 1.** *In FGM, $x_k^+ - x_k$ and $z_{k+1} - z_k$ are parallel[2].*

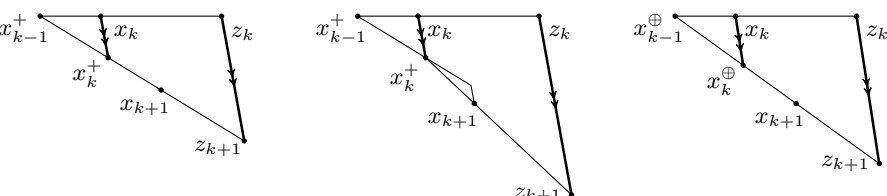

Figure 1: Plane of iteration of FGM (left), OGM (middle), and FISTA-G (right)

Drori, Teboulle, Kim, and Fessler's (**OGM**) [29, 44] has the form:

$$x_{k+1} = x_k^+ + \frac{\theta_k - 1}{\theta_{k+1}}(x_k^+ - x_{k-1}^+) + \frac{\theta_k}{\theta_{k+1}}(x_k^+ - x_k)$$
$$z_k = x_k + (\theta_k - 1)(x_k - x_{k-1}^+) \qquad \text{for } k = 0, 1, \ldots K - 1,$$

where $x_{-1}^+ := x_0$, $z_0 = x_0$, $\theta_0 = 1$, $\theta_{k+1} = \frac{1+\sqrt{1+4\theta_k^2}}{2}$ for $k = 0, 1, \ldots, K - 1$, and $\theta_K = \frac{1+\sqrt{1+8\theta_{K-1}^2}}{2}$. OGM improves upon the rate of FGM [44]. Interestingly, OGM also exhibits a similar geometric structure as depicted in Figure 1 (middle), and the auxiliary sequence, the $z_k$-iterates, plays a similar role in the Lyapunov analysis [63].

**Observation 2.** *In OGM, $x_k^+ - x_k$ and $z_{k+1} - z_k$ are parallel.*

We refer to this geometric structure as the parallel structure. Given an algorithm expressed with momentum and correction terms, one can define the $z_k$-iterates to exhibit the parallel structure:

$$x_{k+1} = x_k^+ + a_{k+1}(x_k^+ - x_{k-1}^+) + b_{k+1}(x_k^+ - x_k) \implies \begin{aligned} x_k &= c_k x_{k-1}^+ + (1 - c_k)z_k \\ z_{k+1} &= z_k - d_k \nabla f(x_k) \end{aligned}$$

---

[2]We define "parallel" to include the degenerate case $x_k^+ - x_k = z_{k+1} - z_k = 0$.

where $a_{k+1}$, $b_{k+1}$, $c_k$, and $d_k$ satisfy an appropriate relationship as shown in Section B of the appendix. The $z_k$-iterates of FGM and OGM have this form. The parallel structure also holds for Güler's accelerated methods [37] for the proximal-point setup and FISTA [11] for the prox-grad setup with analogous definitions of the $z_k$-iterates. The table of Section A.1 of the appendix presents the precise forms.

## 3.2 Parallel structure for OGM-G and FISTA-G

We equivalently write (**OGM-G**) as

$$x_k = \frac{\theta_{k+1}^4}{\theta_k^4} x_{k-1}^+ + \left(1 - \frac{\theta_{k+1}^4}{\theta_k^4}\right) z_k$$

$$z_{k+1} = z_k - \frac{\theta_k}{L} \nabla f(x_k) \qquad \text{for } k = 1, 2, \ldots, K,$$

where $z_0 = x_0$, $z_1 = z_0 - \frac{\theta_0+1}{2L}\nabla f(x_0)$, $\theta_{K+1} = 0$, and $\theta_k = \frac{1+\sqrt{1+4\theta_{k+1}^2}}{2}$ for $k = 1, 2, \ldots, K$, and $\theta_0 = \frac{1+\sqrt{1+8\theta_1^2}}{2}$. The key point is that the auxiliary $z_k$-sequence is defined to exhibit the parallel structure. As a comparison, [47, Equation (43)] provides a different auxiliary sequence for OGM-G that does not exhibit the parallel structure, but this sequence does not lead to a Lyapunov analysis.

Here, we show how the Lyapunov function $U_k$ is obtained from the parallel structure of OGM-G. For $k \geq 1$, combine the cocoercivity inequalities between $(x_{k+1}, x_k)$ and $(x_k, x_K)$ to get

$$0 \geq \frac{1}{\theta_{k+1}^2}\left(f(x_{k+1}) - f(x_k) - \langle \nabla f(x_{k+1}), x_{k+1} - x_k \rangle + \frac{1}{2L}\|\nabla f(x_{k+1}) - \nabla f(x_k)\|^2\right)$$

$$+ \left(\frac{1}{\theta_{k+1}^2} - \frac{1}{\theta_k^2}\right)\left(f(x_k) - f(x_K) - \langle \nabla f(x_k), x_k - x_K \rangle + \frac{1}{2L}\|\nabla f(x_k) - \nabla f(x_K)\|^2\right)$$

$$= \frac{1}{\theta_{k+1}^2}\left(\frac{1}{2L}\|\nabla f(x_K)\|^2 + \frac{1}{2L}\|\nabla f(x_{k+1})\|^2 + f(x_{k+1}) - f(x_K) - \langle \nabla f(x_{k+1}), x_{k+1} - x_k^+ \rangle\right)$$

$$- \frac{1}{\theta_k^2}\left(\frac{1}{2L}\|\nabla f(x_K)\|^2 + \frac{1}{2L}\|\nabla f(x_k)\|^2 + f(x_k) - f(x_K) - \langle \nabla f(x_k), x_k - x_{k-1}^+ \rangle\right)$$

$$\underbrace{- \left\langle \nabla f(x_k), \theta_{k+1}^{-2}x_k^+ - \theta_k^{-2}x_{k-1}^+ - \left(\theta_{k+1}^{-2} - \theta_k^{-2}\right)x_K^+ \right\rangle}_{:=T}.$$

With the following geometric arguments, we analyze the term $T$ and conclude $0 \geq U_{k+1} - U_k$. Let $t \in \mathbb{R}^n$ be the (orthogonal) projection of $x_K^+$ onto the plane of iteration. For $u, v \in \mathbb{R}^n$, define $\overrightarrow{uv} = v - u$. Then

$$\frac{1}{L}T \overset{(i)}{=} \left\langle \overrightarrow{x_k x_k^+}, (\theta_{k+1}^{-2} - \theta_k^{-2})\overrightarrow{tx_k^+} + \theta_k^{-2}\overrightarrow{x_{k-1}^+ x_k^+} \right\rangle$$

$$\overset{(ii)}{=} \left\langle \overrightarrow{x_k x_k^+}, (\theta_{k+1}^{-2} - \theta_k^{-2})(\overrightarrow{tz_{k+1}} - \overrightarrow{z_k z_{k+1}} - \overrightarrow{x_k z_k} + \overrightarrow{x_k x_k^+}) \right.$$

$$\left. + \theta_k^{-2}(\overrightarrow{x_{k-1}^+ x_k} + \overrightarrow{x_k x_k^+}) \right\rangle$$

$$\overset{(iii)}{=} \left\langle \overrightarrow{x_k x_k^+}, (\theta_{k+1}^{-2} - \theta_k^{-2})\overrightarrow{tz_{k+1}} - (\theta_{k+1}^{-2} - \theta_k^{-2})(\theta_k - 1)\overrightarrow{x_k x_k^+} \right.$$

$$\left. - \left(\theta_{k+1}^{-2} - \theta_k^{-2}\right)\overrightarrow{x_k z_k} + (2\theta_k - 1)\theta_{k+1}^{-4}\overrightarrow{x_k z_k} + \theta_k^{-2}\overrightarrow{x_k x_k^+} \right\rangle$$

$$\overset{(iv)}{=} \left\langle \overrightarrow{x_k x_k^+}, \left(\theta_{k+1}^{-2} - \theta_k^{-2}\right)\overrightarrow{tz_{k+1}} + \theta_{k+1}^{-2}(\theta_k - 1)^{-1}\overrightarrow{x_k z_k} \right\rangle$$

$$\overset{(v)}{=} \theta_{k+1}^{-4}\left\langle \overrightarrow{x_k^+ z_{k+1}} - \overrightarrow{x_k z_k}, \overrightarrow{tz_{k+1}} \right\rangle + \theta_{k+1}^{-4}\left\langle \overrightarrow{tz_{k+1}} - \overrightarrow{tz_k}, \overrightarrow{x_k z_k} \right\rangle$$

$$\overset{(vi)}{=} \theta_{k+1}^{-4}\left\langle z_{k+1} - x_k^+, z_{k+1} - x_K^+ \right\rangle - \theta_k^{-4}\left\langle z_k - x_{k-1}^+, z_k - x_K^+ \right\rangle,$$

where (i) follows from the definition of $t$ and the fact that we can replace $x_K^+$ with $t$, the projection of $x_K$ onto the plane of iteration, without affecting the inner products, (ii) from vector addition,

(iii) from the fact that $\overrightarrow{z_k z_{k+1}}$ and $\overrightarrow{x_k x_k^+}$ are parallel and their lengths satisfy $\overrightarrow{z_k z_{k+1}} = \theta_k \overrightarrow{x_k x_k^+}$ and $\overrightarrow{x_{k-1}^+ x_k}$ and $\overrightarrow{x_k z_k}$ are parallel and their lengths satisfy $\overrightarrow{x_{k-1}^+ x_k} = \frac{\theta_k^2 (2\theta_k - 1)}{\theta_{k+1}^4} \overrightarrow{x_k z_k}$, (iv) from the identity $(\theta_{k+1}^{-2} - \theta_k^{-2})(\theta_k - 1) = \theta_k^{-2}$ and $(2\theta_k - 1)\theta_{k+1}^{-4} - (\theta_{k+1}^{-2} - \theta_k^{-2}) = \theta_{k+1}^{-2}(\theta_k - 1)^{-1}$, (v) from distributing the product and substituting $\overrightarrow{x_k x_k^+} = (\theta_k - 1)^{-1}(\overrightarrow{x_k^+ z_{k+1}} - \overrightarrow{x_k z_k})$ (which follows from $\overrightarrow{x_k x_k^+} = \overrightarrow{x_k z_k} + \overrightarrow{z_k z_{k+1}} - \overrightarrow{x_k^+ z_{k+1}} = \overrightarrow{x_k z_k} + \theta_k \overrightarrow{x_k x_k^+} - \overrightarrow{x_k^+ z_{k+1}}$) into the first term and $\overrightarrow{x_k x_k^+} = \theta_k^{-1} \overrightarrow{z_k z_{k+1}} = \theta_k^{-1}\left(\overrightarrow{t z_{k+1}} - \overrightarrow{t z_k}\right)$ into the second term, and the identity $(\theta_{k+1}^{-2} - \theta_k^{-2})(\theta_k - 1)^{-1} = \theta_{k+1}^{-2}(\theta_k - 1)^{-1}\theta_k^{-1} = \theta_{k+1}^{-4}$, and (vi) from cancelling out the cross terms, using $\theta_k^{-4} \overrightarrow{x_{k-1}^+ z_k} = \theta_{k+1}^{-4} \overrightarrow{x_k z_k}$, and by replacing $t$ with $x_K^+$ in the inner products. At this point, the proof is essentially done. The remaining few details of the analysis of OGM-G are presented in Section C of the appendix.

This geometric reasoning naturally extends to the prox-grad setup. Using analogous Lyapunov functions and proof structure, we obtain convergence rates of FISTA-G and the other generalizations presented in Section 3.3. Figure 1 (right) illustrates the parallel structure of FISTA-G with the $z_k$-iterates defined as in Section A.2 of the appendix.

### 3.3   Other generalizations

Using the parallel structure, we find several novel variants of accelerated first-order methods: G-FISTA-G, G-FGM-G, and G-Güler-G. We discuss them in detail in Sections A.2, E, and F of the appendix. Here, we briefly highlight the two special cases that we found most interesting.

We present (**FGM-G**) for the smooth convex setup:

$$x_{k+1} = x_k^+ + \frac{\varphi_{k+1} - \varphi_{k+2}}{\varphi_k - \varphi_{k+1}}(x_k^+ - x_{k-1}^+) \qquad \text{for } k = 0, 1, \ldots, K - 1,$$

where $x_{-1}^+ = x_0$ and $\{\varphi_k\}_{k=0}^{K+1}$ is the same as the $\{\varphi_k\}_{k=0}^{K+1}$ of FISTA-G. We can view FGM-G as a special case of FISTA-G with $g = 0$ or as a special case of G-FGM-G. In any case, FGM-G's final iterate $x_K$ exhibits the rate

$$\|\nabla f(x_K)\|^2 \leq \frac{66L}{(K+2)^2}\left(f(x_0) - f_\star\right).$$

OGM-G uses the "correction term" $\frac{2\theta_{k+1}-1}{2\theta_k-1}(x_k^+ - x_k)$ and the "first-step modification", i.e., $\theta_0$ is defined separately from $\{\theta_k\}_{k=1}^{K-1}$. FGM-G demonstrates that these features are not necessary for achieving the accelerated $\mathcal{O}((f(x_0) - f_\star)/K^2)$ rate. However, the simplification does come at the cost of a worse constant.

We present (**Güler-G**) for the proximal-point setup:

$$x_k = \frac{\theta_{k+1}^4}{\theta_k^4} x_{k-1}^\circ + \left(1 - \frac{\theta_{k+1}^4}{\theta_k^4}\right) z_k$$

$$z_{k+1} = z_k - \theta_k \tilde{\nabla}_{1/\lambda} g(x_k) \qquad \text{for } k = 0, 1, \ldots, K,$$

where $z_0 = x_0$, $\theta_{K+1} = 0$, and $\theta_k = \frac{1 + \sqrt{1 + 4\theta_{k+1}^2}}{2}$ for $k = 0, 1, \ldots, K$.

**Theorem 2.** *Consider* (P) *with* $f = 0$. *Güler-G's final iterate* $x_K$ *exhibits the rate*

$$\left\|\tilde{\nabla}_{1/\lambda} g(x_K)\right\|^2 \leq \frac{4}{\lambda(K+2)^2}(g(x_0) - g_\star).$$

*Furthermore assume $g$ has a minimizer $x_\star$ and define the method Güler+Güler-G as: from a starting point $x_0$ run $K$ iterations of Güler second method [37] and then from the output start Güler-G and run $K$ iterations. Then the final iterate $x_{2K}$ exhibits the rate*

$$\left\|\tilde{\nabla}_{1/\lambda} g(x_{2K})\right\|^2 \leq \frac{4}{\lambda^2(K+2)^4}\|x_0 - x_\star\|^2.$$

# 4 Collinear structure for the strongly convex setup

In this section, we present the *collinear structure*, a geometric structure observed in a wide range of accelerated first-order methods for the strongly convex setup. We then utilize this structure to obtain novel variants of accelerated first-order methods. We specifically consider two strongly convex setups: $f$ is $L$-smooth and $\mu$-strongly convex and $g = 0$ in Section 4.1, while $f = 0$ and $g$ is $\mu$-strongly convex in Section 4.3.

## 4.1 Collinear structure

Nesterov's (**SC-FGM**) [60] has the form:

$$x_{k+1} = x_k^+ + \frac{\sqrt{\kappa} - 1}{\sqrt{\kappa} + 1}(x_k^+ - x_{k-1}^+)$$
$$z_{k+1} = x_{k+1} + \sqrt{\kappa}(x_{k+1} - x_k^+) \qquad \text{for } k = 0, 1, \dots,$$

where $\kappa = \frac{L}{\mu}$ and $x_{-1}^+ := x_0$. Again, the auxiliary $z_k$-iterates reveal the geometric structure and play a key role in the Lyapunov analysis [9, §5.5]. Figure 2 (left) depicts $x_{k-1}^+, x_k, x_k^+, x_k^{++}, x_{k+1}, z_k$, and $z_{k+1}$. These points in $\mathbb{R}^n$ lie on a 2D-plane, which we again call the plane of iteration. Note that $z_k, z_{k+1}$, and $x_k^{++}$ are collinear, i.e., there is a line in $\mathbb{R}^n$ intersecting the three points. (We prove observations 3 and 4 in Section B of the appendix.)

**Observation 3.** *In SC-FGM, $z_k$, $z_{k+1}$, and $x_k^{++}$ are collinear*[3].

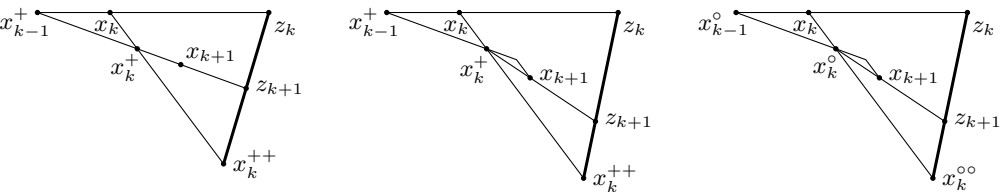

Figure 2: Plane of iteration of SC-FGM (left), TMM (middle), and Proxmal-TMM (right)

Van Scoy, Freeman, and Lynch's (**TMM**) [78], which improves upon SC-FGM, has the form:

$$x_{k+1} = x_k^+ + \frac{(\sqrt{\kappa} - 1)^2}{\sqrt{\kappa}(\sqrt{\kappa} + 1)}(x_k^+ - x_{k-1}^+) + \frac{\sqrt{\kappa} - 1}{\sqrt{\kappa}}(x_k^+ - x_k)$$
$$z_{k+1} = x_{k+1} + \frac{\sqrt{\kappa} - 1}{2}(x_{k+1} - x_k^+) \qquad \text{for } k = 0, 1, \dots,$$

where $x_{-1}^+ := x_0$ and $\kappa = \frac{L}{\mu}$. TMM also exhibits a similar geometric structure as depicted in Figure 2 (middle), and the $z_k$-iterates play a similar role in the Lyapunov analysis [19, Theorem 4.19].

**Observation 4.** *In TMM, $z_k$, $z_{k+1}$, and $x_k^{++}$ are collinear.*

We refer to this geometric structure as the collinear structure. Given an algorithm expressed with momentum and correction terms, one can define the $z_k$-iterates to exhibit the collinear structure:

$$x_{k+1} = x_k^+ + a_k(x_k^+ - x_{k-1}^+) + b_k(x_k^+ - x_k) \quad \Longrightarrow \quad \begin{aligned} x_k &= c_k x_{k-1}^+ + (1 - c_k)z_k \\ z_{k+1} &= d_k x_k^{++} + (1 - d_k)z_k \end{aligned}$$

where $a_k, b_k, c_k$, and $d_k$ satisfy an appropriate relationship as shown in Section B of the appendix. The collinear structure also holds for non-stationary SC-FGM [19, §4.5], SC-OGM [63], ITEM [73], and geometric descent [13, 30]. The table of Section A.1 presents the precise forms.

---

[3] We define "collinear" to include the degenerate case $z_k = z_{k+1} = x_k^{++}$.

## 4.2 Collinear to parallel structure as $\mu \to 0$

We briefly discuss how the parallel structure arises as the limit of the collinear structure as $\mu \to 0$. When $\mu = 0$, the collinear structure is undefined, as $x_k^{++} = x_k - (1/\mu)\nabla f(x_k)$ is undefined. In the limit $\mu \to 0$, however, $x_k^{++}$ diverges to a point of infinity in the plane of iteration, and the collinear structure becomes the parallel structure since $z_k$, $z_{k+1}$, and $x_k^{++}$ are collinear and $x_k$, $x_k^+$, and $x_k^{++}$ are collinear.

There are two possible scenarios. In the first degenerate case, $z_k$ and $z_{k+1}$ diverges to infinity. This is the case with SC-FGM, SC-OGM, and TMM. In the second scenario, $z_k$ and $z_{k+1}$ stay bounded and $x_k^+ - x_k$ and $z_{k+1} - z_k$ become parallel. This is the case with non-stationary[4] SC-FGM and ITEM, which respectively converge to FGM and OGM.

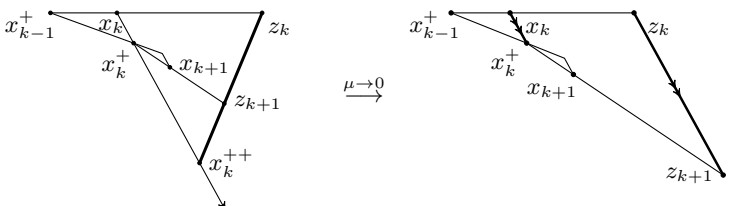

Figure 3: Collinear to parallel structure as $\mu \to 0$ (ITEM (left) $\to$ OGM (right))

## 4.3 Other generalizations

Using the collinear structure, we find two novel methods: Proximal-TMM and Proximal-ITEM. These methods can be viewed as proximal versions of TMM [78] and ITEM [73], and they improve upon the accelerated proximal point method [19, 10]. We provide the proofs in Section G.

We present (**Proximal-TMM**) for the strongly convex proximal-point setup:

$$x_k = \frac{1-\sqrt{q}}{1+\sqrt{q}}x_{k-1}^\circ + \left(1 - \frac{1-\sqrt{q}}{1+\sqrt{q}}\right)z_k$$
$$z_{k+1} = \sqrt{q}x_k^{\circ\circ} + (1-\sqrt{q})z_k \qquad \text{for } k = 0, 1, \ldots,$$

where $q = \frac{\lambda\mu}{\lambda\mu+1}$, $x_k^{\circ\circ} = x_k - \left(\lambda + \frac{1}{\mu}\right)\tilde{\nabla}_{1/\lambda}g(x_k)$, and $x_{-1}^\circ = x_0 = z_0$.

**Theorem 3.** *Consider* (P). *Assume $g$ is $\mu$-strongly convex, $g$ has a minimizer $x_\star$, and $f = 0$. Proximal-TMM's $z_k$-iterates exhibits the rate*

$$\|z_k - x_\star\|^2 \le \frac{2}{\mu}\left(1 - \sqrt{q}\right)^{2k}\left(g(x_0) - g(x_\star)\right).$$

We present (**Proximal-ITEM**) for the strongly convex proximal-point setup:

$$x_k = \gamma_k x_{k-1}^\circ + (1 - \gamma_k)z_k$$
$$z_{k+1} = q\delta_k x_k^{\circ\circ} + (1 - q\delta_k)z_k \qquad \text{for } k = 0, 1, \ldots,$$

where $q = \frac{\lambda\mu}{\lambda\mu+1}$, $x_k^{\circ\circ} = x_k - \left(\lambda + \frac{1}{\mu}\right)\tilde{\nabla}_{1/\lambda}g(x_k)$, $x_{-1}^\circ = x_0 = z_0$, $A_0 = 0$, $A_{k+1} = \frac{(1+q)A_k + 2\left(1 + \sqrt{(1+A_k)(1+qA_k)}\right)}{(1-q)^2}$, $\gamma_k = \frac{A_k}{(1-q)A_{k+1}}$, and $\delta_k = \frac{(1-q)^2 A_{k+1} - (1+q)A_k}{2(1+q+qA_k)}$ for $k = 0, 1, \ldots$.

**Theorem 4.** *Consider* (P). *Assume $g$ is $\mu$-strongly convex, $g$ has a minimizer $x_\star$, and $f = 0$. Proximal-ITEM's $z_k$-iterates exhibits the rate*

$$\|z_k - x_\star\|^2 \le \frac{(1-\sqrt{q})^{2k}}{(1-\sqrt{q})^{2k} + q}\|z_0 - x_\star\|^2.$$

---

[4]We say algorithm is *stationary* if the parameters depend on the iteration [35].

# 5 Experiments

We consider the compressed sensing [25, 15, 14] problems

$$\underset{x\in\mathbb{R}^n}{\text{minimize}} \quad \|Ax - b\|^2 + \lambda \|x\|_1, \qquad \underset{x\in\mathbb{R}^n}{\text{minimize}} \quad \|Ax - b\|^2 + \lambda \|x\|_{\text{nuc}},$$

where $\lambda > 0$, $\|\cdot\|_1$ is the $\ell_1$-norm, $\|\cdot\|_{\text{nuc}}$ is the nuclear norm, and the data $A \in \mathbb{R}^{m \times n}$ and $b \in \mathbb{R}^m$ are generated synthetically. We describe the data generation in Section I of the appendix. Figure 4 presents the comparison of ISTA, FISTA, FPGM-m, FISTA, FISTA-G, and FISTA+FISTA-G. The results indicate that FISTA+FISTA-G is indeed the most effective at reducing the gradient magnitude.

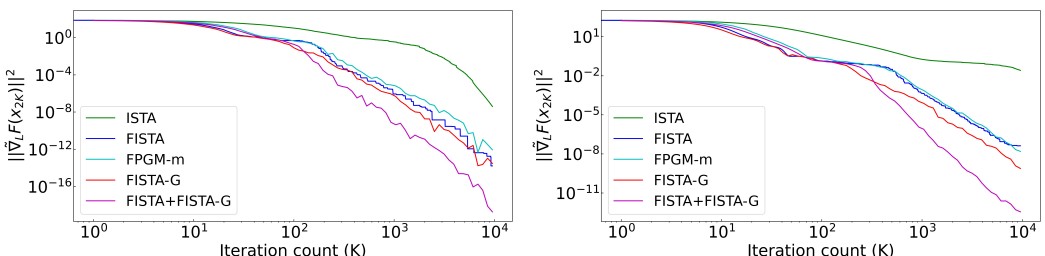

Figure 4: Minimizing the gradient magnitude for compressed sensing problems with $\ell_1$ regularizer (left) and nuclear norm regularizer (right).

# 6 Conclusion

In this work, we identified geometric structures of accelerated first-order methods and utilized them to find novel variants. Specifically, we found that appropriate auxiliary $z_k$-iterates reveal parallel and collinear structures and that these iterates are crucial for the Lyapunov analyses. Among the new methods, we highlight FISTA-G, which, combined with FISTA, achieves a $\mathcal{O}(\|x_0 - x_\star\|^2/K^4)$ rate on the squared gradient norm in the prox-grad setup.

FISTA-G and FISTA+FISTA-G have last-iterate rates, i.e., the bound is on $\|\tilde{\nabla}_L F(x_K)\|^2$ rather than $\min_{k=0,...,K} \|\tilde{\nabla}_L F(x_k)\|^2$, but are not anytime algorithms, i.e., the total iteration count $K$ must be known in advance and intermediate iterates have no guarantees. In contrast, the FGM achieves acceleration on function-value suboptimality with a last-iterate rate as an anytime algorithm. Interestingly, all known algorithms for the smooth convex setup or the prox-grad setup with rate $\mathcal{O}(\|x_0 - x_\star\|^2/K^3)$ or better do not have this property. For example, OGM+OGM-G is not an anytime algorithm but has a last-iterate bound, while FGM is an anytime algorithm with a best-iterate, not a last-iterate, bound. In the prox-grad setup, FPGM-m [45] is not an anytime algorithm but has a last-iterate rate. Whether an anytime algorithm can achieve a last-iterate rate of $\mathcal{O}(\|x_0 - x_\star\|^2/K^3)$ or better is an open problem. On a related note, Diakonikolas and Wang conjecture that a $\mathcal{O}((f(x_0) - f_\star)/K^2)$ rate is impossible to achieve with an anytime algorithm in the smooth convex setup [24, Conjecture 1].

## Acknowledgments and Disclosure of Funding

JL and EKR were supported by the National Research Foundation of Korea (NRF) Grant funded by the Korean Government (MSIP) [No. 2020R1F1A1A01072877], the National Research Foundation of Korea (NRF) Grant funded by the Korean Government (MSIP) [No. 2017R1A5A1015626], and by the Samsung Science and Technology Foundation (Project Number SSTF-BA2101-02). CP was supported by an undergraduate research internship in the first half of the 2021 Seoul National University College of Natural Sciences. We thank Jaewook Suh, Jisun Park, and TaeHo Yoon for providing valuable feedback. We also thank anonymous reviewers for giving thoughtful comments.

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
