# A  Method reference

In this section, we list the methods considered in this work in two forms: a form with momentum and correction terms and a form with the auxiliary iterates. The *momentum and correction terms* of an iteration are loosely defined as

$$x_{k+1} = x_k^+ + \underbrace{a_k(x_k^+ - x_{k-1}^+)}_{\text{momentum term}} + \underbrace{b_k(x_k^+ - x_k)}_{\text{correction term}}.$$

In the proximal-point and prox-grad setup, similar definitions are made with the $x_k^\circ$ and $x_k^\oplus$ terms. One of our main points is that the form with the auxiliary iterates has the advantage of better revealing the parallel and collinear structure, although the form with momentum and correction terms is more commonly presented in the accelerated methods literature. We separate the tables into existing methods and the novel methods we present.

## A.1  Existing Methods

| Method name | With momentum | With auxiliary iterates |
|---|---|---|
| FGM [57] | $$x_{k+1} = x_k^+ + \frac{\theta_k - 1}{\theta_{k+1}}(x_k^+ - x_{k-1}^+)$$ for $k = 0, 1, \ldots$, where $x_{-1}^+ := x_0$, $\theta_0 = 1$, and $\theta_{k+1} = \frac{1+\sqrt{1+4\theta_k^2}}{2}$ for $k = 0, 1, \ldots$ | $$x_k = \frac{\theta_{k-1}^2}{\theta_k^2}x_{k-1}^+ + \left(1 - \frac{\theta_{k-1}^2}{\theta_k^2}\right)z_k$$ $$z_{k+1} = z_k - \theta_k\frac{1}{L}\nabla f(x_k)$$ for $k = 0, 1, \ldots$, where $z_0 = x_0$ and $\theta_{-1} = 0$ |
| OGM [29, 44] | $$x_{k+1} = x_k^+ + \frac{\theta_k - 1}{\theta_{k+1}}(x_k^+ - x_{k-1}^+)$$ $$+ \frac{\theta_k}{\theta_{k+1}}(x_k^+ - x_k)$$ for $k = 0, 1, \ldots, K-1$, where $x_{-1}^+ := x_0$, $\theta_0 = 1$, $\theta_{k+1} = \frac{1+\sqrt{1+4\theta_k^2}}{2}$ for $k = 0, 1, \ldots, K-1$, and $\theta_K = \frac{1+\sqrt{1+8\theta_{K-1}^2}}{2}$ | $$x_k = \frac{\theta_{k-1}^2}{\theta_k^2}x_{k-1}^+ + \left(1 - \frac{\theta_{k-1}^2}{\theta_k^2}\right)z_k$$ $$z_{k+1} = z_k - 2\theta_k\frac{1}{L}\nabla f(x_k)$$ for $k = 0, 1, \ldots, K$, where $z_0 = x_0$ and $\theta_{-1} = 0$ |
| OGM-G [47] | $$x_{k+1} = x_k^+ + \frac{(\theta_k - 1)(2\theta_{k+1} - 1)}{\theta_k(2\theta_k - 1)}(x_k^+ - x_{k-1}^+)$$ $$+ \frac{2\theta_{k+1} - 1}{2\theta_k - 1}(x_k^+ - x_k)$$ for $k = 0, 1, \ldots, K-1$, where $x_{-1}^+ := x_0$, $\theta_K = 1$, $\theta_k = \frac{1+\sqrt{1+4\theta_{k+1}^2}}{2}$ for $k = 1, 2, \ldots, K-1$, and $\theta_0 = \frac{1+\sqrt{1+8\theta_1^2}}{2}$ | $$x_k = \frac{\theta_{k+1}^4}{\theta_k^4}x_{k-1}^+ + \left(1 - \frac{\theta_{k+1}^4}{\theta_k^4}\right)z_k$$ $$z_{k+1} = z_k - \theta_k\frac{1}{L}\nabla f(x_k)$$ for $k = 0, 1, \ldots, K-1$, where $z_0 = x_0$ and $z_1 = z_0 - \frac{\theta_0+1}{2}\frac{1}{L}\nabla f(x_0)$ |
| SC-FGM [60, (2.2.22)] | $$x_{k+1} = x_k^+ + \frac{\sqrt{\kappa} - 1}{\sqrt{\kappa} + 1}(x_k^+ - x_{k-1}^+)$$ for $k = 0, 1, \ldots$, where $\kappa = \frac{L}{\mu}$ and $x_{-1}^+ := x_0$ | $$x_k = \frac{\sqrt{\kappa}}{\sqrt{\kappa} + 1}x_{k-1}^+ + \frac{1}{\sqrt{\kappa} + 1}z_k$$ $$z_{k+1} = \frac{1}{\sqrt{\kappa}}x_k^{++} + \frac{\sqrt{\kappa} - 1}{\sqrt{\kappa}}z_k$$ for $k = 0, 1, \ldots$, where $z_0 = x_0$ |
| non-stationary SC-FGM [19, §4.5] | $$x_{k+1} = x_k^+ + \alpha_k(x_k^+ - x_{k-1}^+)$$ where $\kappa = \frac{L}{\mu}$, $x_{-1}^+ := x_0$, $A_0 = 0$, $A_1 = (1 - \kappa^{-1})^{-1}$, $A_{k+2} = \frac{2A_{k+2}+1+\sqrt{4A_{k+1}+4\kappa^{-1}A_{k+1}^2+1}}{2(1-\kappa^{-1})}$, and $\alpha_k = \frac{(A_{k+2}-A_{k+1})(A_{k+1}(1-\kappa^{-1})-A_k-1)}{A_{k+2}(2\kappa^{-1}A_{k+1}+1)-\kappa^{-1}A_{k+1}^2}$ for $k = 0, 1, \ldots$ | $$x_k = (1 - \gamma_k)x_{k-1}^+ + \gamma_k z_k$$ $$z_{k+1} = \kappa^{-1}\delta_k x_k^{++} + \left(1 - \kappa^{-1}\delta_k\right)z_k$$ for $k = 0, 1, \ldots$ where $z_0 = x_0$, $\gamma_k = \frac{(A_{k+1}-A_k)(1+\kappa^{-1}A_k)}{A_{k+1}+2\kappa^{-1}A_kA_{k+1}-\kappa^{-1}A_k^2}$, and $\delta_k = \frac{A_{k+1}-A_k}{1+\kappa^{-1}A_{k+1}}$ for $k = 0, 1, \ldots$ |

| Method name | With momentum | With auxiliary iterates |
|---|---|---|
| SC-OGM [63] | $$x_{k+1} = x_k^+ + \frac{\kappa - 1}{\sqrt{8\kappa + 1} + 2 + \kappa}(x_k^+ - x_{k-1}^+)$$ $$+ \frac{\kappa - 1}{\sqrt{8\kappa + 1} + 2 + \kappa}(x_k^+ - x_k)$$ for $k = 0, 1, \ldots$, where $\kappa = \frac{L}{\mu}$ and $x_{-1}^+ := x_0$ | $$x_k = \frac{\sqrt{8\kappa + 1} + 3}{2(\sqrt{8\kappa + 1} + 2 + \kappa)}x_{k-1}^+$$ $$+ \frac{\sqrt{8\kappa + 1} + 1 + 2\kappa}{2(\sqrt{8\kappa + 1} + 2 + \kappa)}z_k$$ $$z_{k+1} = \frac{\sqrt{1 + 8\kappa} + 5 - 2\kappa}{\sqrt{1 + 8\kappa} + 3}x_k^{++} + \frac{2\kappa - 2}{\sqrt{1 + 8\kappa} + 3}z_k$$ for $k = 0, 1, \ldots$, where $z_0 = x_0$ |
| TMM [78] | $$x_{k+1} = x_k^+ + \frac{(\sqrt{\kappa} - 1)^2}{\sqrt{\kappa}(\sqrt{\kappa} + 1)}(x_k^+ - x_{k-1}^+)$$ $$+ \frac{\sqrt{\kappa} - 1}{\sqrt{\kappa}}(x_k^+ - x_k)$$ for $k = 0, 1, \ldots$, where $x_{-1}^+ := x_0$ and $\kappa = \frac{L}{\mu}$ | $$x_k = \frac{\sqrt{\kappa} - 1}{\sqrt{\kappa} + 1}x_{k-1}^+ + \frac{2}{\sqrt{\kappa} + 1}z_k$$ $$z_{k+1} = \frac{1}{\sqrt{\kappa}}x_k^{++} + \frac{\sqrt{\kappa} - 1}{\sqrt{\kappa}}z_k$$ for $k = 0, 1, \ldots$, where $z_0 = x_0$ |
| Geometric descent [13, 30] | $$z_0 = x_0^{++}, R_0^2 = \left(1 - \frac{1}{\kappa}\right)\frac{\|\nabla f(x_0)\|^2}{\mu^2}$$ $$\lambda_{k+1} = \arg\min_{\lambda \in \mathbb{R}} f((1 - \lambda)c_t + \lambda x_k^+)$$ $$x_{k+1} = (1 - \lambda_{k+1})z_k + \lambda_{k+1}x_k^+$$ If $\frac{|\nabla f(x_k)\|^2}{\mu^2} < \frac{R_k^2}{2}$, $$z_{k+1} = x_{k+1}^{++}$$ $$R_{k+1}^2 = \frac{\|\nabla f(x_{k+1})\|^2/\mu^2}{1 - \kappa^{-1}}.$$ If $\frac{\|\nabla f(x_k)\|^2}{\mu^2} \geq \frac{R_k^2}{2}$, $$z_{k+1} = (1 - \frac{R_k^2 + \|x_{k+1} - z_k\|^2}{2\|x_{k+1}^{++} - z_k\|^2})z_k + \frac{R_k^2 + \|x_{k+1} - z_k\|^2}{2\|x_{k+1}^{++} - z_k\|^2}x_{k+1}^{++}$$ $$R_{k+1}^2 = R_k^2 - \frac{\|\nabla f(x_k)\|^2}{\mu^2 \kappa} - \left(\frac{R_k^2 + \|x_{k+1} - z_k\|^2}{2\|x_{k+1}^{++} - z_k\|^2}\right)^2$$ | |
| ITEM [73] | $$x_{k+1} = x_k^+ + \alpha_k(x_k^+ - x_{k-1}^+) + \beta_k(x_k^+ - x_k)$$ $k = 0, 1, \ldots$, where $\kappa = \frac{L}{\mu}, x_{-1}^+ := x_0, A_0 = 0,$ $A_1 = (1 - \kappa^{-1})^{-1},$ $A_{k+2} = \frac{(1+\kappa^{-1})A_{k+1} + 2(1 + \sqrt{(1 + A_{k+1})(1 + \kappa^{-1}A_{k+1})}}{(1 - \kappa^{-1})^2},$ $\alpha_k = \frac{(2(1+\kappa^{-1}) + \kappa^{-1}(3 + \kappa^{-1})A_k + (1 - \kappa^{-1})^2\kappa^{-1}A_{k+1})((1 - \kappa^{-1})A_{k+2} - A_{k+1})A_k}{2(1 - \kappa^{-1})(1 + \kappa^{-1} + \kappa^{-1}A_k)((1 - \kappa^{-1})A_{k+1} - A_k)A_{k+2}},$ and $\beta_k = \frac{(\kappa^{-1}A_k^2 + 2(1 - \kappa^{-1})A_{k+1} + (1 - \kappa^{-1})\kappa^{-1}A_kA_{k+1})((1 - \kappa^{-1})A_{k+2} - A_{k+1})}{2(1 + \kappa^{-1} + \kappa^{-1}A_k)((1 - \kappa^{-1})A_{k+1} - A_k)A_{k+2}}$ for $k = 0, 1, \ldots$ | $$x_k = \gamma_k x_{k-1}^+ + (1 - \gamma_k)z_k$$ $$z_{k+1} = \kappa^{-1}\delta_k x_k^{++} + (1 - \kappa^{-1}\delta_k)z_k$$ for $k = 0, 1, \ldots$, where $z_0 = x_0, A_0 = 0, \kappa = \frac{L}{\mu}$, $\gamma_k = \frac{A_k}{(1 - \kappa^{-1})A_{k+1}}$ and $\delta_k = \frac{(1 - \kappa^{-1})^2A_{k+1} - (1 + \kappa^{-1})A_k}{1 + \kappa^{-1} + \kappa^{-1}A_k}$ for $k = 0, 1, \ldots$ |
| ISTA [20] | $$x_{k+1} = x_k^{\oplus} \qquad \text{for } k = 0, 1, \ldots$$ | |

| Method name | With momentum | With auxiliary iterates |
|---|---|---|
| FISTA [11] | $$x_{k+1} = x_k^\oplus + \frac{\theta_k - 1}{\theta_{k+1}}(x_k^\oplus - x_{k-1}^\oplus)$$ for $k = 0, 1, \ldots$, where $x_{-1}^\oplus := x_0$, $\theta_0 = 1$, and $\theta_{k+1} = \frac{1+\sqrt{1+4\theta_k^2}}{2}$ for $k = 0, 1, \ldots$ | $$x_k = \frac{\theta_{k-1}^2}{\theta_k^2}x_{k-1}^\oplus + \left(1 - \frac{\theta_{k-1}^2}{\theta_k^2}\right)z_k$$ $$z_{k+1} = z_k - \theta_k \frac{1}{L}\tilde{\nabla}_L F(x_k)$$ for $k = 0, 1, \ldots$, where $z_0 = x_0$ |
| FPGM-m [45] | $$x_{k+1} = x_k^\oplus + \frac{\theta_k - 1}{\theta_{k+1}}(x_k^\oplus - x_{k-1}^\oplus) \qquad \text{for } 0 \le k \le m-1$$ $$x_{k+1} = x_k^\oplus \qquad \text{for } m \le k \le K$$ where $x_{-1}^\oplus := x_0$, $\theta_0 = 1$, and $\theta_{k+1} = \frac{1+\sqrt{1+4\theta_k^2}}{2}$ for $k = 0, 1, \ldots, m-1$ | |
| Güler 1 [37] | $$x_{k+1} = x_k^\circ + \frac{\theta_k - 1}{\theta_{k+1}}(x_k^\circ - x_{k-1}^\circ)$$ for $k = 0, 1, \ldots$, where $x_{-1}^\circ := x_0$, $\theta_0 = 1$, and $\theta_{k+1} = \frac{1+\sqrt{1+4\theta_k^2}}{2}$ for $k = 0, 1, \ldots$ | $$x_k = \frac{\theta_{k-1}^2}{\theta_k^2}x_{k-1}^\circ + \left(1 - \frac{\theta_{k-1}^2}{\theta_k^2}\right)z_k$$ $$z_{k+1} = z_k - \theta_k \tilde{\nabla}_{1/\lambda}g(x_k)$$ for $k = 0, 1, \ldots$, where $z_0 = x_0$ and $\theta_{-1} = 0$ |
| Güler 2 [37] | $$x_{k+1} = x_k^\circ + \frac{\theta_k - 1}{\theta_{k+1}}(x_k^\circ - x_{k-1}^\circ) + \frac{\theta_k}{\theta_{k+1}}(x_k^\circ - x_k)$$ for $k = 0, 1, \ldots$, where $x_{-1}^\circ := x_0$, $\theta_0 = 1$, and $\theta_{k+1} = \frac{1+\sqrt{1+4\theta_k^2}}{2}$ for $k = 0, 1, \ldots$ | $$x_k = \frac{\theta_{k-1}^2}{\theta_k^2}x_{k-1}^\circ + \left(1 - \frac{\theta_{k-1}^2}{\theta_k^2}\right)z_k$$ $$z_{k+1} = z_k - 2\theta_k \tilde{\nabla}_{1/\lambda}g(x_k)$$ for $k = 0, 1, \ldots$, where $z_0 = x_0$ and $\theta_{-1} = 0$ |

## A.2 Novel methods

| Method name | With momentum | With auxiliary iterates |
|---|---|---|
| FISTA-G | $$x_{k+1} = x_k^{\oplus} + \frac{\varphi_{k+1} - \varphi_{k+2}}{\varphi_k - \varphi_{k+1}}(x_k^{\oplus} - x_{k-1}^{\oplus})$$ for $k = 0, 1, \ldots, K-1$, where $x_{-1}^{\oplus} := x_0$, $\varphi_{K+1} = 0$, $\varphi_K = 1$, and $\varphi_k = \frac{\varphi_{k+2}^2 - \varphi_{k+1}\varphi_{k+2} + 2\varphi_{k+1}^2 + (\varphi_{k+1} - \varphi_{k+2})\sqrt{\varphi_{k+2}^2 + 3\varphi_{k+1}^2}}{\varphi_{k+1} + \varphi_{k+2}}$ for $k = 0, 1, \ldots, K-1$ | $$x_k = \frac{\varphi_{k+1}}{\varphi_k}x_{k-1}^{\oplus} + \left(1 - \frac{\varphi_{k+1}}{\varphi_k}\right)z_k$$ $$z_{k+1} = z_k - \frac{\varphi_k}{\varphi_k - \varphi_{k+1}}\frac{1}{L}\tilde{\nabla}_L F(x_k)$$ for $k = 0, 1, \ldots, K$, where $z_0 = x_0$ |
| G-FISTA-G | $$x_{k+1} = x_k^{\oplus} + \frac{\varphi_{k+1} - \varphi_{k+2}}{\varphi_k - \varphi_{k+1}}(x_k^{\oplus} - x_{k-1}^{\oplus})$$ $$+ \frac{\varphi_{k+1} - \varphi_{k+2}}{\varphi_{k+1}}\left(\tau_k\varphi_k - \tau_{k+1}\varphi_{k+1} - \frac{\varphi_k}{\varphi_k - \varphi_{k+1}}\right)(x_k^{\oplus} - x_k)$$ for $k = 0, 1, \ldots, K-1$, where $x_{-1}^{\oplus} := x_0$, $\tau_K = \varphi_K = 1$, $\varphi_{K+1} = 0$, and $\{\varphi_k\}_{k=0}^{K-1}$ and the nondecreasing nonnegative sequence $\{\tau_k\}_{k=0}^{K-1}$ satisfying $\tau_k\varphi_k - \tau_{k+1}\varphi_{k+1} = \varphi_{k+1}(\tau_{k+1} - \tau_k) + 1$, and $(\tau_k\varphi_k - \tau_{k+1}\varphi_{k+1})(\tau_{k+1} - \tau_k) - \frac{\tau_{k+1}}{2} \leq 0$ for $k = 0, 1, \ldots, K-1$ | $$x_k = \frac{\varphi_{k+1}}{\varphi_k}x_{k-1}^{\oplus} + \left(1 - \frac{\varphi_{k+1}}{\varphi_k}\right)z_k$$ $$z_{k+1} = z_k - (\tau_k\varphi_k - \tau_{k+1}\varphi_{k+1})\frac{1}{L}\tilde{\nabla}_L F(x_k)$$ for $k = 0, 1, \ldots, K$, where $z_0 = x_0$ |
| FGM-G | $$x_{k+1} = x_k^{+} + \frac{\varphi_{k+1} - \varphi_{k+2}}{\varphi_k - \varphi_{k+1}}(x_k^{+} - x_{k-1}^{+})$$ for $k = 0, 1, \ldots, K-1$, where $x_{-1}^{+} := x_0$, $\varphi_{K+1} = 0$, $\varphi_K = 1$, and $\varphi_k = \frac{\varphi_{k+2}^2 - \varphi_{k+1}\varphi_{k+2} + 2\varphi_{k+1}^2 + (\varphi_{k+1} - \varphi_{k+2})\sqrt{\varphi_{k+2}^2 + 3\varphi_{k+1}^2}}{\varphi_{k+1} + \varphi_{k+2}}$ for $k = 0, 1, \ldots, K-1$ | $$x_k = \frac{\varphi_{k+1}}{\varphi_k}x_{k-1}^{+} + \left(1 - \frac{\varphi_{k+1}}{\varphi_k}\right)z_k$$ $$z_{k+1} = z_k - \frac{\varphi_k}{\varphi_k - \varphi_{k+1}}\frac{1}{L}\nabla f(x_k)$$ for $k = 0, 1, \ldots, K$, where $z_0 = x_0$ |
| G-FGM-G | $$x_{k+1} = x_k^{+} + \frac{\varphi_{k+1} - \varphi_{k+2}}{\varphi_k - \varphi_{k+1}}(x_k^{+} - x_{k-1}^{+})$$ $$+ \frac{\varphi_{k+1} - \varphi_{k+2}}{\varphi_{k+1}}\left(\tau_k\varphi_k - \tau_{k+1}\varphi_{k+1} - \frac{\varphi_k}{\varphi_k - \varphi_{k+1}}\right)(x_k^{+} - x_k)$$ for $k = 0, 1, \ldots, K-1$ where $x_{-1}^{+} := x_0$, $\tau_K = \varphi_K = 1$, $\varphi_{K+1} = 0$, and $\{\varphi_k\}_{k=0}^{K-1}$ and the nondecreasing nonnegative sequence $\{\tau_k\}_{k=0}^{K-1}$ satisfying $\tau_k\varphi_k - \tau_{k+1}\varphi_{k+1} = \varphi_{k+1}(\tau_{k+1} - \tau_k) + 1$ and $(\tau_k\varphi_k - \tau_{k+1}\varphi_{k+1})(\tau_{k+1} - \tau_k) - \tau_{k+1} \leq 0$ for $k = 0, 1, \ldots, K-1$ | $$x_k = \frac{\varphi_{k+1}}{\varphi_k}x_{k-1}^{+} + \left(1 - \frac{\varphi_{k+1}}{\varphi_k}\right)z_k$$ $$z_{k+1} = z_k - (\tau_k\varphi_k - \tau_{k+1}\varphi_{k+1})\frac{1}{L}\nabla f(x_k)$$ for $k = 0, 1, \ldots, K$, where $z_0 = x_0$ |
| Güler-G | $$x_{k+1} = x_k^{\circ} + \frac{(\theta_k - 1)(2\theta_{k+1} - 1)}{\theta_k(2\theta_k - 1)}(x_k^{\circ} - x_{k-1}^{\circ})$$ $$+ \frac{2\theta_{k+1} - 1}{2\theta_k - 1}(x_k^{\circ} - x_k)$$ for $k = 0, 1, \ldots, K-1$, where $x_{-1}^{\circ} := x_0$, $\theta_K = 1$, and $\theta_k = \frac{1 + \sqrt{1 + 4\theta_{k+1}^2}}{2}$ for $k = 0, 1, \ldots, K-1$ | $$x_k = \frac{\theta_{k+1}^4}{\theta_k^4}x_{k-1}^{\circ} + \left(1 - \frac{\theta_{k+1}^4}{\theta_k^4}\right)z_k$$ $$z_{k+1} = z_k - \theta_k\tilde{\nabla}_{1/\lambda}g(x_k)$$ for $k = 0, 1, \ldots, K$, where $z_0 = x_0$ and $\theta_{K+1} = 0$ |

| Method name | With momentum | With auxiliary iterates |
|---|---|---|
| G-Güler-G | $x_{k+1} = x_k^\circ + \frac{\varphi_{k+1} - \varphi_{k+2}}{\varphi_k - \varphi_{k+1}}(x_k^\circ - x_{k-1}^\circ)$ $+ \frac{\varphi_{k+1} - \varphi_{k+2}}{\varphi_{k+1}}\left(\tau_k \varphi_k - \tau_{k+1}\varphi_{k+1} - \frac{\varphi_k}{\varphi_k - \varphi_{k+1}}\right)(x_k^\circ - x_k)$ for $k = 0, 1, \ldots, K-1$ where $x_{-1}^\circ := x_0$, $\tau_K = \varphi_K = 1$, $\varphi_{K+1} = 0$, and $\{\varphi_k\}_{k=0}^{K-1}$ and the nondecreasing nonnegative sequence $\{\tau_k\}_{k=0}^{K-1}$ satisfying $\tau_k \varphi_k - \tau_{k+1}\varphi_{k+1} = \varphi_{k+1}(\tau_{k+1} - \tau_k) + 1$ and $(\tau_k \varphi_k - \tau_{k+1}\varphi_{k+1})(\tau_{k+1} - \tau_k) - \tau_{k+1} \leq 0$ for $k = 0, 1, \ldots, K-1$ | $x_k = \frac{\varphi_{k+1}}{\varphi_k}x_{k-1}^\circ + \left(1 - \frac{\varphi_{k+1}}{\varphi_k}\right)z_k$ $z_{k+1} = z_k - (\tau_k \varphi_k - \tau_{k+1}\varphi_{k+1})\frac{1}{L}\tilde{\nabla}_{1/\lambda}g(x_k)$ for $k = 0, 1, \ldots, K$, where $z_0 = x_0$ |
| Proximal -TMM | $x_{k+1} = x_k^\circ + \frac{(\sqrt{q} - 1)^2}{\sqrt{q} + 1}(x_k^\circ - x_{k-1}^\circ)$ $+ (1 - \sqrt{q})(x_k^\circ - x_k)$ for $k = 0, 1, \ldots$, where $x_{-1}^\circ := x_0$ and $q = \frac{\lambda\mu}{\lambda\mu+1}$ | $x_k = \frac{1 - \sqrt{q}}{1 + \sqrt{q}}x_{k-1}^\circ + \left(1 - \frac{1 - \sqrt{q}}{1 + \sqrt{q}}\right)z_k$ $z_{k+1} = \sqrt{q}x_k^{\circ\circ} + (1 - \sqrt{q})z_k$ for $k = 0, 1, \ldots$, where $x_k^{\circ\circ} = x_k - \left(\lambda + \frac{1}{\mu}\right)\tilde{\nabla}_{1/\lambda}g(x_k)$, and $x_{-1}^\circ = x_0 = z_0$ for $k = 0, 1, \ldots$ |
| Proximal -ITEM | $x_{k+1} = x_k^\circ + \alpha_k(x_k^\circ - x_{k-1}^\circ) + \beta_k(x_k^\circ - x_k)$ for $k = 0, 1, \ldots$, where $q = \frac{\lambda\mu}{\lambda\mu+1}$, $x_{-1}^\circ := x_0$, $A_0 = 0$, $A_1 = (1 - q)^{-1}$, $A_{k+2} = \frac{(1+q)A_{k+1} + 2(1 + \sqrt{(1+A_{k+1})(1+qA_{k+1})})}{(1-q)^2}$, $\alpha_k = \frac{(2(1+q)+q(3+q)A_k+(1-q)^2qA_{k+1})((1-q)A_{k+2}-A_{k+1})A_k}{2(1-q)(1+q+qA_k)((1-q)A_{k+1}-A_k)A_{k+2}}$, and $\beta_k = \frac{(qA_k^2+2(1-q)A_{k+1}+(1-q)qA_kA_{k+1})((1-q)A_{k+2}-A_{k+1})}{2(1+q+qA_k)((1-q)A_{k+1}-A_k)A_{k+2}}$ for $k = 0, 1, \ldots$ | $x_k = \gamma_k x_{k-1}^\circ + (1 - \gamma_k)z_k$ $z_{k+1} = q\delta_k x_k^{\circ\circ} + (1 - q\delta_k)z_k$ for $k = 0, 1, \ldots$, where $z_0 = x_0$, $x_k^{\circ\circ} = x_k - \left(\lambda + \frac{1}{\mu}\right)\tilde{\nabla}_{1/\lambda}g(x_k)$, $\gamma_k = \frac{A_k}{(1-q)A_{k+1}}$, and $\delta_k = \frac{(1-q)^2 A_{k+1} - (1+q)A_k}{2(1+q+qA_k)}$ for $k = 0, 1, \ldots$ |

## B Omitted proofs of geometric observation and form of algorithm

In this section, we formally establish the basic geometric claims made in the main body.

First, we state parallel lemma (left) and Menelaus's lemma (right), which are classical results in Euclidean geometry:

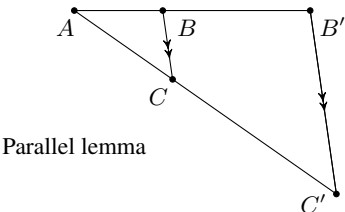

| Parallel lemma | Menelaus's lemma |

$\overline{BC} \parallel \overline{B'C'}$ if and only if $\frac{\overline{AB}}{\overline{BB'}} = \frac{\overline{AC}}{\overline{CC'}}$
 $\qquad$ $A', B', C'$ is on line if and only if $\frac{\overline{A'B}}{\overline{AA'}} \cdot \frac{\overline{B'C}}{\overline{BB'}} \cdot \frac{\overline{C'A}}{\overline{CC'}} = 1$

### B.1 Omitted proofs of observations

*Proof of Observation 1.* Figure 1 (left) depicts the plane of iteration of FGM. In the plane of iteration of FGM,

$$\frac{\|x_k - x_{k-1}^+\|}{\|z_k - x_k\|} = \frac{1}{\theta_k - 1} = \frac{1}{\frac{\theta_k-1}{\theta_{k+1}} + \frac{(\theta_k-1)(\theta_{k+1}-1)}{\theta_{k+1}}} = \frac{\|x_k^+ - x_{k-1}^+\|}{\|x_{k+1} - x_k^+\| + \|z_{k+1} - x_{k+1}\|} = \frac{\|x_k^+ - x_{k-1}^+\|}{\|z_{k+1} - x_k^+\|}$$

by definition of $z_k, x_{k+1}, z_{k+1}$. Then the result comes from parallel lemma. $\qquad\square$

*Proof of Observation 2.* Figure 1 (middle) depicts the plane of iteration of OGM. By extending $\overline{x_{k-1}^+ x_k^+}$ and defining new point $B$ that meets with $\overleftrightarrow{z_k z_{k+1}}$, observation can also be shown by parallel lemma. $\qquad\square$

*Proof of Observation 3.* Figure 2 (left) depicts the plane of iteration of SC-FGM. Apply Menelaus's lemma for $\triangle x_{k-1}^+ x_k x_k^+$ and $\overline{z_k z_{k+1} x_k^{++}}$, that

$$\frac{\|z_k - x_{k-1}^+\|}{\|z_k - x_k\|} \cdot \frac{\|z_{k+1} - x_k^+\|}{\|z_{k+1} - x_{k-1}^+\|} \cdot \frac{\|x_k^{++} - x_k\|}{\|x_k^{++} - x_k^+\|} = \frac{\sqrt{\kappa} + 1}{\sqrt{\kappa}} \cdot \frac{\sqrt{\kappa} - 1}{\sqrt{\kappa}} \cdot \frac{\frac{1}{\mu}}{\frac{1}{\mu} - \frac{1}{L}} = 1.$$

$\qquad\square$

*Proof of Observation 4.* Figure 2 (middle) depicts the plane of iteration of TMM. By extending $\overline{x_k^+ x_{k+1}}$ and defining new point $Q$ that meets with $\overleftrightarrow{x_{k-1}^+ x_k}$, Observation 4 can be shown by Menelaus's lemma for $\triangle Q x_k x_k^+$ and $\overline{z_k z_{k+1} x_k^{++}}$. $\qquad\square$

### B.2 Parallel structure from momentum-based iteration

**Lemma 1.** *An iteration of the form*

$$x_{k+1} = x_k^+ + \frac{a_k - 1}{a_{k+1}}(x_k^+ - x_{k-1}^+) + b_{k+1}(x_k^+ - x_k)$$

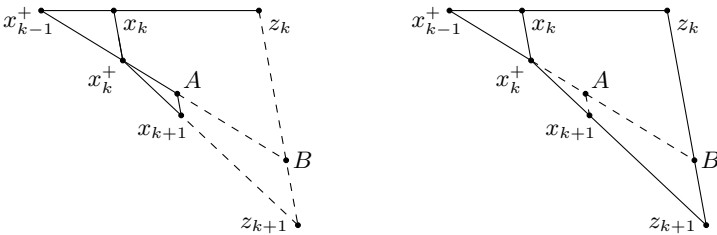

Figure 5: Lemma 1 and Lemma 2

*for $k = 0, 1, \ldots, K - 1$, where $1 \le a_0$, $1 < a_k$ for $k = 1, 2, \ldots, K - 1$, and $1 \le a_K$, can be equivalently expressed as*

$$x_k = \frac{\varphi_{k-1}}{\varphi_k} x_{k-1}^+ + \left(1 - \frac{\varphi_{k-1}}{\varphi_k}\right) z_k$$

$$z_{k+1} = z_k - \frac{a_k + a_{k+1} b_{k+1}}{L} \nabla f(x_k)$$

*for $k = 0, 1, \ldots, K$, where $0 = \varphi_{-1}$, $0 < \varphi_k$, $\frac{a_k - 1}{a_k} = \frac{\varphi_{k-1}}{\varphi_k}$ for $k = 1, 2, \ldots, K$, and $\varphi_K \le \infty$. (If $\varphi_K = \infty$, we define $\varphi_{K-1}/\varphi_K = 0$.)*

*Proof.* First, suppose $x_k$ is not a minimizer which implies $\nabla f(x_k) \ne 0$ and neither $x_{k-1}^+$, $x_k$, $z_k$ are not the same. From first iteration of algorithm with auxiliary iterates, we know $x_{k-1}^+, x_k, z_k$ are collinear. Set $A$ on the $\overleftrightarrow{x_{k-1}^+ x_k^+}$ that $\overline{A x_{k+1}} \parallel \overline{x_k x_k^+}$. Let $B$ on the $\overleftrightarrow{x_k^+ A}$ that $\overline{z_k B} \parallel \overline{x_k x_k^+}$. Lastly, we set $z_{k+1} := \overleftrightarrow{z_k B} \cap \overleftrightarrow{x_k^+ x_{k+1}}$. Then, the condition for parallel term style is satisfied. We will show that the formula above also holds.

Since $\overline{x_k x_k^+} \parallel \overline{z_k B}$, parallel lemma indicates that

$$\frac{\|z_k - x_k\|}{\|x_k - x_{k-1}^+\|} = \frac{\|B - x_k^+\|}{\|x_k^+ - x_{k-1}^+\|} = \frac{\varphi_{k-1}}{\varphi_k - \varphi_{k-1}}$$

Since $\overline{A x_{k+1}} \parallel \overline{B z_{k+1}}$, parallel lemma indicates that

$$\frac{\|z_{k+1} - x_k^+\|}{\|x_{k+1} - x_k^+\|} = \frac{\|B - x_k^+\|}{\|A - x_k^+\|} = \frac{\varphi_{k+1}}{\varphi_{k+1} - \varphi_k}$$

Then,

$$\frac{\|A - x_k^+\|}{\|x_k^+ - x_{k-1}^+\|} = \frac{a_k - 1}{a_{k+1}} = \frac{\varphi_{k+1} - \varphi_k}{\varphi_{k+1}} \cdot \frac{\varphi_{k-1}}{\varphi_k - \varphi_{k-1}}$$

and this relation holds if $a_{k+1} = \frac{\varphi_{k+1}}{\varphi_{k+1} - \varphi_k} \iff \frac{a_{k+1} - 1}{a_{k+1}} = \frac{\varphi_k}{\varphi_{k+1}}$. (This strong condition is for easy Lyapunov analysis).

Lastly, by parallel lemma and previous condition,

$$z_{k+1} - B = \frac{\|z_{k+1} - x_k^+\|}{\|x_{k+1} - x_k^+\|}(x_{k+1} - A) = a_{k+1} b_{k+1}(x_k^+ - x_k)$$

since $\|x_{k+1} - A\| = b_{k+1}\|x_k^+ - x_k\|$ and

$$B - z_k = \frac{\|z_k - x_{k-1}^+\|}{\|x_k - x_{k-1}^+\|}(x_k^+ - x_k) = a_k(x_k^+ - x_k),$$

which indicates

$$z_{k+1} = z_k - (a_k + a_{k+1}b_{k+1})\frac{1}{L}\nabla f(x_k).$$

If $x_k$ is a minimizer which implies $\nabla f(x_k) = 0$ and $z_{k+1} - z_k = x_k^+ - x_k = 0$, this is degenerate case. In this case, proof is trivial. $\qquad\square$

**Lemma 2.** *An iteration of the form*

$$x_k = \frac{\varphi_{k-1}}{\varphi_k}x_{k-1}^+ + \left(1 - \frac{\varphi_{k-1}}{\varphi_k}\right)z_k$$

$$z_{k+1} = z_k - \frac{\phi_k}{L}\nabla f(x_k)$$

*for $k = 0, 1, \ldots, K$, where $\{\varphi_k\}_{k=-1}^K$ is a nonnegative increasing sequence, can be equivalently expressed as*

$$x_{k+1} = x_k^+ + \frac{\varphi_{k+1} - \varphi_k}{\varphi_{k+1}} \cdot \frac{\varphi_{k-1}}{\varphi_k - \varphi_{k-1}}(x_k^+ - x_{k-1}^+) + \frac{\varphi_{k+1} - \varphi_k}{\varphi_{k+1}}\left(\phi_k - \frac{\varphi_k}{\varphi_k - \varphi_{k-1}}\right)(x_k^+ - x_k)$$

*for $k = 0, 1, \ldots, K$.*

*Proof.* Suppose $x_k$ is not a minimizer which implies $\nabla f(x_k) \neq 0$. Set $A$ on the $\overleftrightarrow{x_{k-1}^+ x_k^+}$ that $\overline{Ax_{k+1}} \parallel \overline{x_k x_k^+}$. Let $B$ on the $\overleftrightarrow{x_{k-1}^+ x_k^+} \cap \overline{z_k z_{k+1}}$. Since $\overline{x_k x_k^+} \parallel \overline{z_k B}$ and $\overline{Ax_{k+1}} \parallel \overline{Bz_{k+1}}$,

$$A - x_k^+ = (B - x_k^+) - (B - A) = (B - x_k^+) - \frac{\varphi_k}{\varphi_{k+1}}(B - x_k^+)$$

$$= \frac{\varphi_{k+1} - \varphi_k}{\varphi_{k+1}}(B - x_k^+) = \frac{\varphi_{k+1} - \varphi_k}{\varphi_{k+1}} \cdot \frac{\varphi_{k-1}}{\varphi_k - \varphi_{k-1}}(x_k^+ - x_{k-1}^+).$$

In addition, since $\overline{x_k x_k^+} \parallel \overline{z_k B}$ and $\overline{Ax_{k+1}} \parallel \overline{Bz_{k+1}}$,

$$x_{k+1} - A = \frac{\varphi_{k+1} - \varphi_k}{\varphi_{k+1}}(z_{k+1} - B) = \frac{\varphi_{k+1} - \varphi_k}{\varphi_{k+1}}((z_{k+1} - z_k) - (B - z_k))$$

$$= \frac{\varphi_{k+1} - \varphi_k}{\varphi_{k+1}}\left(\phi_k(x_k^+ - x_k) - \frac{\varphi_k}{\varphi_k - \varphi_{k-1}}(x_k^+ - x_k)\right)$$

$$= \frac{\varphi_{k+1} - \varphi_k}{\varphi_{k+1}}\left(\phi_k - \frac{\varphi_k}{\varphi_k - \varphi_{k-1}}\right)(x_k^+ - x_k).$$

If $x_k$ is a minimizer which implies $\nabla f(x_k) = 0$ and $z_{k+1} - z_k = x_k^+ = x_k = 0$, this is degenerate case. In this case, proof is trivial. $\qquad\square$

By Lemmas 1 and 2, there is a correspondence between the two algorithm forms.

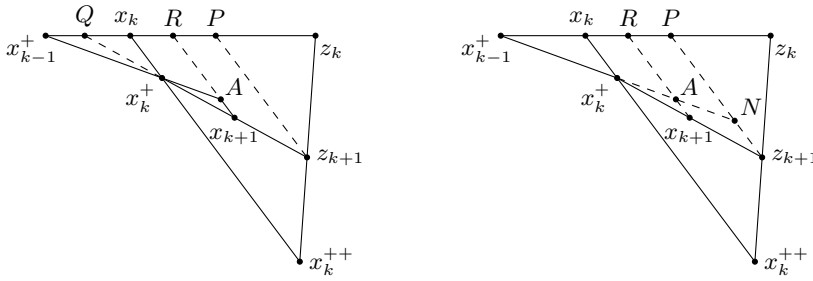

Figure 6: Lemma 3 and Lemma 4

## B.3 Collinear structure from momentum-based iteration

**Lemma 3.** *An iteration of the form*

$$x_{k+1} = x_k^+ + a_k(x_k^+ - x_{k-1}^+) + b_k(x_k^+ - x_k),$$

*where $0 < a_k$ and $0 \le b_k$, for $k = 0, 1, \ldots$. can be equivalently expressed as*

$$x_k = (1 - \varphi_k)x_{k-1}^+ + \varphi_k z_k$$

$$z_{k+1} = \left(1 - \frac{a_k \varphi_k}{(1 - \varphi_k)\varphi_{k+1}}\right) x_k^{++} + \frac{a_k \varphi_k}{(1 - \varphi_k)\varphi_{k+1}} z_k$$

*for $k = 0, 1, \ldots$, where $\varphi_{k+1} = (a_k + b_k) \cdot \frac{\mu}{L - \mu} + \frac{a_k \varphi_k}{1 - \varphi_k} \cdot \frac{L}{L - \mu}$, provided that $1 > \varphi_k > 0$ for $k = 0, 1, \ldots$.*

*Proof.* Suppose $x_k$ is not a minimizer which implies $\nabla f(x_k) \ne 0$ and neither $x_{k-1}^+, x_k, z_k$ are not the same. From first iteration of algorithm with auxiliary iterates, we know $x_{k-1}^+, x_k, z_k$ are collinear. We inductively set $z_k$, and we set $Q := \overleftrightarrow{x_{k-1}^+ x_k} \cap \overleftrightarrow{x_k^+ x_{k+1}}$ and $z_{k+1} := \overleftrightarrow{x_k^+ x_{k+1}} \cap \overleftrightarrow{x_k^{++} z_k}$. Set $A$ on the $\overleftrightarrow{x_{k-1}^+ x_k^+}$ that $\overline{Ax_{k+1}}$ is parallel to $\overline{x_k x_k^+}$. Set $R := \overleftrightarrow{x_{k-1}^+ z_k} \cap \overline{Ax_{k+1}}$. Set $P$ on the $\overleftrightarrow{x_{k-1}^+ z_k}$ that $\overline{Pz_{k+1}}$ is parallel to $\overline{x_k x_k^+}$.

By parallel lemma,

$$\frac{\|x_k^+ - x_k\|}{\|A - R\|} = \frac{\|x_k^+ - x_{k-1}^+\|}{\|A - x_{k-1}^+\|} = \frac{1}{1 + a_k}$$

and

$$\frac{\|x_k^+ - x_k\|}{\|x_{k+1} - R\|} = \frac{\|x_k^+ - x_k\|}{\|x_{k+1} - A\| + \|A - R\|} = \frac{1}{1 + a_k + b_k}.$$

Then, we have

$$\frac{\|x_k - Q\|}{\|R - x_k\|} = \frac{\|x_k^+ - x_k\|}{\|R - x_{k+1}\| - \|x_k^+ - x_k\|} = \frac{1}{a_k + b_k}$$

and

$$\frac{\|x_k - x_{k-1}^+\|}{\|x_k - Q\|} = \frac{\|x_k - x_{k-1}^+\|}{\|R - x_k\|} \cdot \frac{\|R - x_k\|}{\|x_k - Q\|} = \frac{\|x_k - x_{k-1}^+\|}{\|R - x_k\|} \cdot \frac{\|A - x_k^+\|}{\|x_k^+ - x_{k-1}^+\|} = \frac{a_k + b_k}{a_k}.$$

Also parallel lemma implies

$$\frac{\|R - x_k\|}{\|P - R\|} = \frac{\|x_{k+1} - x_k^+\|}{\|z_{k+1} - x_{k+1}\|} = \frac{\varphi_{k+1}}{1 - \varphi_{k+1}}.$$

Applying Menelaus's lemma to $\triangle Q x_k x_k^+$ and $\overline{z_k z_{k+1} x_k^{++}}$,

$$\frac{\|z_{k+1} - x_k^+\|}{\|z_{k+1} - Q\|} \cdot \frac{\|x_k^{++} - x_k\|}{\|x_k^{++} - x_k^+\|} \cdot \frac{\|z_k - Q\|}{\|z_k - x_k\|} = 1.$$

Using $\frac{\|z_{k+1} - x_k^+\|}{\|z_{k+1} - Q\|} = \frac{\|P - x_k\|}{\|P - Q\|}$, $\frac{\|x_k^{++} - z_{k+1}\|}{\|z_{k+1} - z_k\|} = \frac{\|z_k - P\|}{\|P - x_k\|}$, and previous formula, we get

$$\varphi_{k+1} = (a_k + b_k) \cdot \frac{\mu}{L - \mu} + \frac{a_k \varphi_k}{1 - \varphi_k} \cdot \frac{L}{L - \mu}.$$

Furthermore, parallel lemma and $\frac{\|z_k - x_k\|}{\|x_k - x_{k-1}^+\|} = \frac{1 - \varphi_k}{\varphi_k}$ implies

$$\frac{\|x_k - x_{k-1}^+\|}{\|R - x_k\|} = \frac{\|x_k^+ - x_{k-1}^+\|}{\|A - x_k^+\|} = \frac{1}{a_k}$$

and

$$\frac{\|P - R\|}{\|z_k - P\|} = \frac{a_k \frac{1 - \varphi_{k+1}}{\varphi_{k+1}}}{\frac{1 - \varphi_k}{\varphi_k} - a_k \left(\frac{1 - \varphi_{k+1}}{\varphi_{k+1}} + 1\right)}.$$

Therefore, we have

$$\frac{\|x_k^{++} - z_{k+1}\|}{\|z_{k+1} - z_k\|} = \frac{\|P - x_k\|}{\|z_k - P\|} = \frac{\frac{a_k}{\varphi_{k+1}}}{\frac{1 - \varphi_k}{\varphi_k} - \frac{a_k}{\varphi_{k+1}}} = \frac{a_k \varphi_k}{(1 - \varphi_k)\varphi_{k+1} - a_k \varphi_k}.$$

If $x_k$ is a minimizer which implies $\nabla f(x_k) = 0$ and $z_{k+1} = z_k = x_k^{++}$, this is degenerate case. In this case, proof is trivial. $\square$

**Lemma 4.** *An iteration of the form*

$$x_k = (1 - \varphi_k)x_{k-1}^+ + \varphi_k z_k$$
$$z_{k+1} = (1 - \phi_k)x_k^{++} + \phi_k z_k,$$

*where $0 < \varphi_k$ and $0 < \phi_k$, for $k = 0, 1, \ldots$ can be equivalently expressed as*

$$x_{k+1} = \frac{(1 - \varphi_k)\varphi_{k+1}\phi_k}{\varphi_k}(x_k^+ - x_{k-1}^+) + \frac{\varphi_{k+1}((\kappa - 1)(1 - \phi_k)\varphi_k - \phi_k)}{\varphi_k}(x_k^+ - x_k)$$

*for $k = 0, 1, \ldots$.*

*Proof.* Suppose $x_k$ is not a minimizer which implies $\nabla f(x_k) \neq 0$. Set $A$ on the $\overleftrightarrow{x_{k-1}^+ x_k^+}$ that $\overline{Ax_{k+1}} \parallel \overline{x_k x_k^+}$. Set $P$ on the $\overleftrightarrow{x_{k-1}^+ z_k}$ that $\overline{Pz_{k+1}} \parallel \overline{x_k x_k^+}$. Set $N := \overleftrightarrow{x_k^+ A} \cap \overleftrightarrow{Pz_{k+1}}$. Lastly, set $R := \overleftrightarrow{x_{k-1}^+ z_k} \cap \overleftrightarrow{Ax_{k+1}}$.

By parallel lemma, we have

$$\frac{\|P - x_k\|}{\|z_k - P\|} = \frac{\|x_k^{++} - z_{k+1}\|}{\|z_{k+1} - z_k\|} = \frac{\phi_k}{1 - \phi_k}$$

and

$$\frac{\|R - x_k\|}{\|P - R\|} = \frac{\|x_{k+1} - x_k^+\|}{\|z_{k+1} - x_{k+1}\|} = \frac{\varphi_{k+1}}{1 - \varphi_{k+1}}.$$

Also $\frac{\|x_k - x_{k-1}^+\|}{\|z_k - x_k\|} = \frac{\varphi_k}{1 - \varphi_k}$ and previous formula implies

$$\frac{\|x_k - x_{k-1}^+\|}{\|P - x_k\|} = \frac{\varphi_k}{\phi_k(1 - \varphi_k)}$$

and

$$\frac{\|x_k - x_{k-1}^+\|}{\|R - x_k\|} = \frac{\varphi_k}{(1 - \varphi_k)\varphi_{k+1}\phi_k}$$

Furthermore, we get

$$A - x_k^+ = \frac{\|A - x_k^+\|}{\|x_k^+ - x_{k-1}^+\|}(x_k^+ - x_{k-1}^+) = \frac{\|R - x_k\|}{\|x_k - x_{k-1}^+\|}(x_k^+ - x_{k-1}^+) = \frac{(1 - \varphi_k)\varphi_{k+1}\phi_k}{\varphi_k}(x_k^+ - x_{k-1}^+).$$

By parallel lemma, we have

$$\frac{\|x_k^{++} - x_k\|}{\|z_{k+1} - P\|} = \frac{\|x_k^{++} - z_k\|}{\|z_{k+1} - z_k\|} = \frac{1}{1 - \phi_k}$$

and

$$\frac{\|N - P\|}{\|x_k^+ - x_k\|} = \frac{\|P - x_{k-1}^+\|}{\|x_k - x_{k-1}^+\|} = \frac{\varphi_k + (1 - \varphi_k)\phi_k}{\varphi_k}.$$

Using $\frac{\|x_k^+ - x_k\|}{\|x_k^{++} - x_k\|} = \frac{1}{\kappa}$,

$$\frac{\|z_{k+1} - P\|}{\|x_k^+ - x_k\|} = \kappa(1 - \phi_k)$$

and previous formula implies

$$\frac{\|z_{k+1} - N\|}{\|x_k^+ - x_k\|} = \frac{\|z_{k+1} - P\| - \|N - P\|}{\|x_k^+ - x_k\|} = \frac{(\kappa - 1)(1 - \phi_k)\varphi_k - \phi_k}{\varphi_k}.$$

Finally, we get

$$x_{k+1} - A = \frac{\|z_{k+1} - N\|}{\|x_k^+ - x_k\|}\frac{\|x_{k+1} - A\|}{\|z_{k+1} - N\|}(x_k^+ - x_k) = \frac{\varphi_{k+1}((\kappa - 1)(1 - \phi_k)\varphi_k - \phi_k)}{\varphi_k}(x_k^+ - x_k).$$

If $x_k$ is a minimizer which implies $\nabla f(x_k) = 0$ and $z_{k+1} = z_k = x_k^{++}$, this is degenerate case. In this case, proof is trivial. $\square$

By Lemmas 3 and 4, there is a correspondence between the two algorithm forms.

## C  OGM-G analysis

Using Lemma 1, we can write OGM-G [47] as

$$x_k = \frac{\theta_{k+1}^4}{\theta_k^4}x_{k-1}^+ + \left(1 - \frac{\theta_{k+1}^4}{\theta_k^4}\right)z_k$$

$$z_{k+1} = z_k - \frac{\theta_k}{L}\nabla f(x_k),$$

where $z_0 = x_0$ and $z_1 = z_0 - \frac{\theta_0 + 1}{2L}\nabla f(x_0)$ for $k = 1, 2, \ldots K$.

**Theorem 5.** *Consider* (P) *with* $g = 0$. *OGM-G's* $x_K$ *exhibits the rate*

$$\|\nabla f(x_K)\|^2 \leq \frac{2L}{\theta_0^2}(f(x_0) - f_\star) \leq \frac{4L}{(K + 1)^2}(f(x_0) - f_\star).$$

*Proof.* For $k = 1, 2, \ldots, K$, define

$$U_k = \frac{1}{\theta_k^2} \left( \frac{1}{2L} \|\nabla f(x_K)\|^2 + \frac{1}{2L} \|\nabla f(x_k)\|^2 + f(x_k) - f(x_K) - \left\langle \nabla f(x_k), x_k - x_{k-1}^+ \right\rangle \right)$$
$$+ \frac{L}{\theta_k^4} \left\langle z_k - x_{k-1}^+, z_k - x_K^+ \right\rangle$$

and

$$U_0 = \frac{2}{\theta_0^2} \left( \frac{1}{2L} \|\nabla f(x_K)\|^2 + f(x_0) - f(x_K) \right).$$

We can show that $\{U_k\}_{k=0}^K$ is nonincreasing. Using $\frac{1}{2L} \|\nabla f(x_K)\|^2 \leq f(x_K) - f(x_K^+) \leq f(x_K) - f_\star$, which follows from $L$-smoothness, we conclude with the rate

$$\frac{1}{L} \|\nabla f(x_K)\|^2 = U_K \leq U_0 \leq \frac{2}{\theta_0^2} \left( f(x_0) - f_\star \right)$$

and the bound $\theta_0 \geq \frac{K+1}{\sqrt{2}}$ [47, Theorem 6.1]. Now, we complete the proof by showing that $\{U_k\}_{k=0}^K$ is nonincreasing. As we already showed $U_1 \geq U_2 \geq \cdots \geq U_K$ in Section 3.2, all that remains is to show $U_0 \geq U_1$:

$U_0 - U_1$

$$= -\frac{1}{\theta_1^2} f(x_1) + \frac{2}{\theta_0^2} f(x_0) + \left( \frac{1}{\theta_1^2} - \frac{2}{\theta_0^2} \right) f(x_K) - \frac{1}{\theta_1^2} \frac{1}{2L} \|\nabla f(x_1)\|^2 - \left( \frac{1}{\theta_1^2} - \frac{2}{\theta_0^2} \right) \frac{1}{2L} \|\nabla f(x_K)\|^2$$
$$+ \frac{1}{\theta_1^2} \left\langle \nabla f(x_1), x_1 - x_0^+ \right\rangle - \frac{L}{\theta_1^4} \left\langle z_1 - x_0^+, z_1 - x_K^+ \right\rangle$$

$$= -\frac{1}{\theta_1^2} \left( f(x_1) - f(x_0) - \left\langle \nabla f(x_1), x_1 - x_0^+ \right\rangle + \frac{1}{2L} \|\nabla f(x_1)\|^2 + \frac{1}{2L} \|\nabla f(x_0)\|^2 \right)$$
$$- \left( \frac{1}{\theta_1^2} - \frac{2}{\theta_0^2} \right) \left( f(x_0) - f(x_K) - \left\langle \nabla f(x_0), x_0 - x_K^+ \right\rangle + \frac{1}{2L} \|\nabla f(x_0)\|^2 + \frac{1}{2L} \|\nabla f(x_K)\|^2 \right)$$
$$+ \left( \frac{1}{\theta_1^2} - \frac{1}{\theta_0^2} \right) \frac{1}{L} \|\nabla f(x_0)\|^2 - \left( \frac{1}{\theta_1^2} - \frac{2}{\theta_0^2} \right) \left\langle \nabla f(x_0), x_0 - x_K^+ \right\rangle - \frac{L}{\theta_1^4} \left\langle z_1 - x_0^+, z_1 - x_K^+ \right\rangle$$

$$\geq \left( \frac{1}{\theta_1^2} - \frac{1}{\theta_0^2} \right) \frac{1}{L} \|\nabla f(x_0)\|^2 - \left( \frac{1}{\theta_1^2} - \frac{2}{\theta_0^2} \right) \left\langle \nabla f(x_0), x_0 - x_K^+ \right\rangle - \frac{L}{\theta_1^4} \left\langle z_1 - x_0^+, z_1 - x_K^+ \right\rangle$$

$$= \frac{\theta_0 + 1}{\theta_0^2 (\theta_0 - 1)} \frac{1}{L} \|\nabla f(x_0)\|^2 - \frac{1}{\theta_1^2 \theta_0} \left\langle \nabla f(x_0), x_0 - x_K^+ \right\rangle - \frac{L}{\theta_1^4} \left\langle z_1 - x_0^+, z_1 - x_K^+ \right\rangle$$

$$= 0$$

where the inequality follows from the cocoercivity inequalities. $\qquad \square$

## D    Several preliminary inequalities

**Lemma 5** ([60, (2.1.11)]). *If $f : \mathbb{R}^n \to \mathbb{R}$ is convex and $L$-smooth, then*

$$f(x) - f(y) + \left\langle \nabla f(x), y - x \right\rangle + \frac{1}{2L} \|\nabla f(x) - \nabla f(y)\|^2 \leq 0 \qquad \forall x, y \in \mathbb{R}^n,$$

$$f(y) \leq f(x) + \left\langle \nabla f(x), y - x \right\rangle + \frac{L}{2} \|x - y\|^2 \qquad \forall x, y \in \mathbb{R}^n.$$

**Lemma 6.** *If $g \colon \mathbb{R}^n \to \mathbb{R} \cup \{\infty\}$ is $\mu$-strongly convex, then for all $u \in \partial g(x)$,*

$$g(x) + \langle u, y - x \rangle + \frac{\mu}{2} \|x - y\|^2 \leq g(y) \qquad \forall x, y \in \mathbb{R}^n.$$

**Lemma 7** ([11, lemma 2.2]). *Consider* (P) *in the prox-grad setup. Then for some $u \in \partial g(x^\oplus)$,*

$$\tilde{\nabla}_L F(x) = \nabla f(x) + u \qquad \forall x \in \mathbb{R}^n.$$

*Proof.* Optimality condition for strongly convex function implies that there exist $u \in \partial g(x^\oplus)$ such that $\nabla f(x) + u + L(x^\oplus - x) = 0$. $\qquad\square$

**Lemma 8** ([45, (2.8)]). *Consider* (P) *in the prox-grad setup. Then for some $v \in \partial F(x^\oplus)$,*

$$\|v\| \leq 2 \left\| \tilde{\nabla}_L F(x) \right\| \qquad \forall x \in \mathbb{R}^n.$$

*Proof.* By Lemma 7, $\tilde{\nabla}_L F(x) = \nabla f(x) + u$ for some $u \in \partial g(x^\oplus)$. And there exist $v \in \partial F(x^\oplus)$ such that $v = \nabla f(x^\oplus) + u$. Thus we have

$$\|v\| \leq \|\nabla f(x^\oplus) - \nabla f(x)\| + \|\nabla f(x) + u\| \tag{1}$$

$$\leq \|L(x - x^\oplus)\| + \|\tilde{\nabla}_L F(x)\| \tag{2}$$

$$= 2 \left\| \tilde{\nabla}_L F(x) \right\|. \tag{3}$$

(1) follows from triangle inequality and (2) follows from $L$-smoothness of $f$, and (3) follows from the definition of $\tilde{\nabla}_L F(x)$. $\qquad\square$

**Lemma 9** ([59, Theorem 1]). *Consider* (P) *in the prox-grad setup. Then*

$$\frac{1}{2L} \left\| \tilde{\nabla}_L F(x) \right\|^2 \leq F(x) - F(x^\oplus) \qquad \forall x \in \mathbb{R}^n.$$

*Proof.* By Lemma 7, for some $u \in \partial g(x^\oplus)$, we have

$$F(x^\oplus) \leq f(x) + \langle \nabla f(x), x^\oplus - x \rangle + \frac{L}{2} \left\| x^\oplus - x \right\|^2 + g(x^\oplus) \tag{4}$$

$$\leq f(x) + \langle L(x - x^\oplus) - u, x^\oplus - x \rangle + \frac{L}{2} \left\| x^\oplus - x \right\|^2 + g(x^\oplus) \tag{5}$$

$$= f(x) + g(x^\oplus) + \langle u, x - x^\oplus \rangle - \frac{L}{2} \left\| x^\oplus - x \right\|^2$$

$$\leq F(x) - \frac{L}{2} \left\| x^\oplus - x \right\|^2 = F(x) - \frac{1}{2L} \left\| \tilde{\nabla}_L F(x) \right\|^2. \tag{6}$$

(4) follows from $L$-smoothness of $f$, (5) follows from the definition of $\tilde{\nabla}_L F(x)$, and (6) follows from convexity of $g$. $\qquad\square$

**Lemma 10** ([11, lemma 2.3]). *Consider* (P) *in the prox-grad setup. Then*

$$\frac{1}{2L} \left\| \tilde{\nabla}_L F(y) \right\|^2 - \left\langle y - x, \tilde{\nabla}_L F(y) \right\rangle \leq F(x) - F(y^\oplus) \qquad \forall x, y \in \mathbb{R}^n.$$

*Proof.* By $L$-smoothness of $f$, we have

$$F(y^{\oplus}) \leq f(y) + \langle \nabla f(y), y^{\oplus} - y \rangle + \frac{L}{2} \left\| y^{\oplus} - y \right\|^2 + g(y^{\oplus}).$$

Using convexity of $f$ and $g$ and Lemma 7, for some $u \in \partial g(y^{\oplus})$, we have

$$f(y) + \langle \nabla f(y), x - y \rangle \leq f(x)$$
$$g(y^{\oplus}) + \langle u, x - y^{\oplus} \rangle \leq g(x).$$

Summing above three inequality, we obtain the lemma. $\qquad\square$

## E    Omitted proofs of Section 2

We present (**G-FISTA-G**) for the prox-grad setup:

$$x_k = \frac{\varphi_{k+1}}{\varphi_k} x_{k-1}^{\oplus} + \left(1 - \frac{\varphi_{k+1}}{\varphi_k}\right) z_k$$
$$z_{k+1} = z_k - \frac{\tau_k \varphi_k - \tau_{k+1} \varphi_{k+1}}{L} \tilde{\nabla}_L F(x_k)$$

for $k = 0, 1, \ldots, K$, where $z_0 = x_0$, $L$ is smoothness constant of $f$, and the nonnegative sequence $\{\varphi_k\}_{k=0}^{K+1}$ and the nondecreasing nonnegative sequence $\{\tau_k\}_{k=0}^{K}$ satisfy $\varphi_{K+1} = 0$, $\varphi_K = \tau_K = 1$, and

$$\tau_k \varphi_k - \tau_{k+1} \varphi_{k+1} = \varphi_{k+1}(\tau_{k+1} - \tau_k) + 1, \qquad (\tau_k \varphi_k - \tau_{k+1} \varphi_{k+1})(\tau_{k+1} - \tau_k) \leq \frac{\tau_{k+1}}{2}$$

for $k = 0, 1, \ldots, K - 1$.

**Theorem 6.** *Consider* (P). *G-FISTA-G's $x_K$ exhibits the rate*

$$\left\| \tilde{\nabla}_L F(x_K) \right\|^2 \leq 2L\tau_0 \left( F(x_0) - F_\star \right).$$

*Proof.* For $k = 0, 1, \ldots, K$, define

$$U_k = \tau_k \left( \frac{1}{2L} \left\| \tilde{\nabla}_L F(x_k) \right\|^2 + F(x_k^{\oplus}) - F(x_K^{\oplus}) - \left\langle \tilde{\nabla}_L F(x_k), x_k - x_{k-1}^{\oplus} \right\rangle \right)$$
$$+ \frac{L}{\varphi_k} \left\langle z_k - x_{k-1}^{\oplus}, z_k - x_K^{\oplus} \right\rangle.$$

(Note that $z_K = x_K$.) By plugging in the definitions and performing direct calculations, we get

$$U_K = \frac{1}{2L} \left\| \tilde{\nabla}_L F(x_k) \right\|^2 \qquad \text{and} \qquad U_0 = \tau_0 \left( \frac{1}{2L} \left\| \tilde{\nabla}_L F(x_0) \right\|^2 + F(x_0^{\oplus}) - F(x_K^{\oplus}) \right).$$

We can show that $\{U_k\}_{k=0}^K$ is nonincreasing. Using Lemma 9, we conclude the rate with

$$\frac{1}{2L} \left\| \tilde{\nabla}_L F(x_k) \right\|^2 = U_K \leq U_0 \leq \tau_0 \left( F(x_0) - F(x_K^{\oplus}) \right) \leq \tau_0 \left( F(x_0) - F_\star \right).$$

Now we complete the proof by showing that $\{U_k\}_{k=0}^K$ is nonincreasing. For $k = 0, 1, \ldots, K-1$, we have

$$0 \geq \tau_{k+1}\left(F(x_{k+1}^{\oplus}) - F(x_k^{\oplus}) - \left\langle \tilde{\nabla}_L F(x_{k+1}), x_{k+1} - x_k^{\oplus} \right\rangle + \frac{1}{2L}\left\|\tilde{\nabla}_L F(x_{k+1})\right\|^2\right)$$

$$+ (\tau_{k+1} - \tau_k)\left(F(x_k^{\oplus}) - F(x_K^{\oplus}) - \left\langle \tilde{\nabla}_L F(x_k), x_k - x_K^{\oplus}\right\rangle + \frac{1}{2L}\left\|\tilde{\nabla}_L F(x_k)\right\|^2\right)$$

$$= \tau_{k+1}\left(\frac{1}{2L}\left\|\tilde{\nabla}_L F(x_{k+1})\right\|^2 + F(x_{k+1}^{\oplus}) - F(x_K^{\oplus}) - \left\langle \tilde{\nabla}_L F(x_{k+1}), x_{k+1} - x_k^{\oplus}\right\rangle\right)$$

$$- \tau_k\left(\frac{1}{2L}\left\|\tilde{\nabla}_L F(x_k)\right\|^2 + F(x_k^{\oplus}) - F(x_K^{\oplus}) - \left\langle \tilde{\nabla}_L F(x_k), x_k - x_{k-1}^{\oplus}\right\rangle\right)$$

$$\underbrace{- \left\langle \tilde{\nabla}_L F(x_k), \tau_{k+1}x_k^{\oplus} - \tau_k x_{k-1}^{\oplus} - (\tau_{k+1} - \tau_k)x_K^{\oplus}\right\rangle - \tau_{k+1}\frac{1}{2L}\left\|\tilde{\nabla}_L F(x_k)\right\|^2}_{:=T},$$

where the inequality follows from the Lemma 10. Finally, we analyze $T$ with the following geometric argument. Let $t \in \mathbb{R}^n$ be the projection of $x_K^{\oplus}$ onto the plane of iteration. Then,

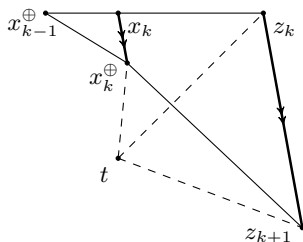

Figure 7: Plane of iteration of G-FISTA-G

$$\frac{1}{L}T \stackrel{\text{(i)}}{=} \left\langle \overrightarrow{x_k x_k^{\oplus}}, (\tau_{k+1} - \tau_k)\overrightarrow{t x_k^{\oplus}} + \tau_k \overrightarrow{x_{k-1}^{\oplus} x_k^{\oplus}} - \frac{\tau_{k+1}}{2}\overrightarrow{x_k x_k^{\oplus}}\right\rangle$$

$$\stackrel{\text{(ii)}}{=} \left\langle \overrightarrow{x_k x_k^{\oplus}}, (\tau_{k+1} - \tau_k)\left(\overrightarrow{t z_{k+1}} - \overrightarrow{z_k z_{k+1}} - \overrightarrow{x_k z_k} + \overrightarrow{x_k x_k^{\oplus}}\right) + \tau_k\left(\overrightarrow{x_{k-1}^{\oplus} x_k} + \overrightarrow{x_k x_k^{\oplus}}\right) - \frac{\tau_{k+1}}{2}\overrightarrow{x_k x_k^{\oplus}}\right\rangle$$

$$\stackrel{\text{(iii)}}{=} \left\langle \overrightarrow{x_k x_k^{\oplus}}, (\tau_{k+1} - \tau_k)\overrightarrow{t z_{k+1}} - (\tau_{k+1} - \tau_k)(\tau_k \varphi_k - \tau_{k+1}\varphi_{k+1} - 1)\overrightarrow{x_k x_k^{\oplus}} \right.$$

$$\left. + \tau_k \overrightarrow{x_k x_k^{\oplus}} - \frac{\tau_{k+1}}{2}\overrightarrow{x_k x_k^{\oplus}} - (\tau_{k+1} - \tau_k)\overrightarrow{x_k z_k} + \tau_k\left(\frac{\varphi_k}{\varphi_{k+1}} - 1\right)\overrightarrow{x_k z_k}\right\rangle$$

$$\stackrel{\text{(iv)}}{\geq} \left\langle \overrightarrow{x_k x_k^{\oplus}}, (\tau_{k+1} - \tau_k)\overrightarrow{t z_{k+1}} + \frac{\tau_k \varphi_k - \tau_{k+1}\varphi_{k+1}}{\varphi_{k+1}}\overrightarrow{x_k z_k}\right\rangle$$

$$\stackrel{\text{(v)}}{=} \frac{1}{\varphi_{k+1}}\left\langle \overrightarrow{x_k^{\oplus} z_{k+1}} - \overrightarrow{x_k z_k}, \overrightarrow{t z_{k+1}}\right\rangle + \frac{1}{\varphi_{k+1}}\left\langle \overrightarrow{t z_{k+1}} - \overrightarrow{t z_k}, \overrightarrow{x_k z_k}\right\rangle$$

$$\stackrel{\text{(vi)}}{=} \frac{1}{\varphi_{k+1}}\left\langle z_{k+1} - x_k^{\oplus}, z_{k+1} - x_K^{\oplus}\right\rangle - \frac{1}{\varphi_k}\left\langle z_k - x_{k-1}^{\oplus}, z_k - x_K^{\oplus}\right\rangle$$

where (i) follows from the definition of $t$ and the fact that we can replace $x_K^\oplus$ with $t$, the projection of $x_K$ onto the plane of iteration, without affecting the inner products, (ii) from vector addition, (iii) from the fact that $\overrightarrow{x_k x_k^\oplus}$ and $\overrightarrow{z_k z_{k+1}}$ are parallel and their lengths satisfy $(\tau_k \varphi_k - \tau_{k+1}\varphi_{k+1})\overrightarrow{x_k x_k^\oplus} = \overrightarrow{z_k z_{k+1}}$ and $\overrightarrow{x_{k-1}^\oplus x_k}$ and $\overrightarrow{x_k z_k}$ are parallel and their lengths satisfy $\left(\frac{\varphi_k}{\varphi_{k+1}} - 1\right)\overrightarrow{x_k z_k} = \overrightarrow{x_{k-1}^\oplus x_k}$, (iv) from vector addition and

$$\frac{\tau_{k+1}}{2} - (\tau_k \varphi_k - \tau_{k+1}\varphi_{k+1})(\tau_{k+1} - \tau_k) \geq 0, \tag{7}$$

(v) from distributing the product and substituting $\overrightarrow{x_k x_k^\oplus} = (\tau_k \varphi_k - \tau_{k+1}\varphi_{k+1} - 1)^{-1}\left(\overrightarrow{x_k^\oplus z_{k+1}} - \overrightarrow{x_k z_k}\right) = (\varphi_{k+1}(\tau_{k+1} - \tau_k))^{-1}\left(\overrightarrow{x_k^\oplus z_{k+1}} - \overrightarrow{x_k z_k}\right)$ into the first term and $\overrightarrow{x_k x_k^\oplus} = (\tau_k \varphi_k - \tau_{k+1}\varphi_{k+1})^{-1}\overrightarrow{z_k z_{k+1}} = (\tau_k \varphi_k - \tau_{k+1}\varphi_{k+1})^{-1}\left(\overrightarrow{t z_{k+1}} - \overrightarrow{t z_k}\right)$ into the second term, and (vi) from cancelling out the cross terms, using $\varphi_k^{-1}\overrightarrow{x_{k-1}^\oplus z_k} = \varphi_{k+1}^{-1}\overrightarrow{x_k z_k}$, and by replacing $t$ with $x_K^\oplus$ in the inner products. In (v), we also used

$$\tau_k \varphi_k - \tau_{k+1}\varphi_{k+1} = \varphi_{k+1}(\tau_{k+1} - \tau_k) + 1. \tag{8}$$

Thus we conclude $U_{k+1} \leq U_k$ for $k = 0, 1, 2, \ldots, K - 1$. $\qquad\square$

*Proof of Theorem 1.* The conclusion of Theorem 1 follows from plugging FISTA-G's $\varphi_k$ and $\tau_k$ into Theorem 6. If $\tau_k = \frac{2\varphi_{k-1}}{(\varphi_{k-1} - \varphi_k)^2}$, we can check condition (8), condition (7), and

$$\varphi_k \tau_k - \tau_{k+1}\varphi_{k+1} = \varphi_{k+1}(\tau_{k+1} - \tau_k) + 1 = \frac{\varphi_k}{\varphi_k - \varphi_{k+1}}.$$

Then using Lemma 2, we get the iteration of the form of FISTA-G. Furthermore, by Lemma 8, $\|v\| \leq 2\left\|\tilde{\nabla}_L F(x)\right\|$ for some $v \in \partial F(x^\oplus)$. Thus we have

$$\min\left\|\partial F(x_K^\oplus)\right\|^2 \leq 4\left\|\tilde{\nabla}_L F(x_K)\right\|^2 \leq \frac{264L}{(K+2)^2}\left(F(x_0) - F_\star\right).$$

Finally, it remains to show $\tau_0 \leq \frac{33}{(K+2)^2}$ in the setup of Theorem 1.

First, $\tau_k \varphi_k - \tau_{k+1}\varphi_{k+1} = \varphi_{k+1}(\tau_{k+1} - \tau_k) + 1$ and $(\tau_k \varphi_k - \tau_{k+1}\varphi_{k+1})(\tau_{k+1} - \tau_k) - \frac{\tau_{k+1}}{2} = 0$ implies

$$\varphi_{k+1} = \frac{1}{\tau_{k+1} - \tau_k}\left(\frac{\tau_{k+1}}{2(\tau_{k+1} - \tau_k)} - 1\right).$$

By substitution and direct calculation, we get

$$a_{k+1} = a_k + \frac{(a_k - a_{k-1})a_k}{\sqrt{a_k^2 - a_k a_{k-1} + a_{k-1}^2}},$$

where $\tau_k = \frac{1}{a_{K-k}}$. This is equivalent to

$$\frac{a_{k+1}}{a_k} = 1 + \frac{(a_k - a_{k-1})}{\sqrt{(a_k - a_{k-1})^2 + a_k a_{k-1}}} \iff \frac{a_{k+1}}{a_k} = 1 + \frac{1}{\sqrt{1 + \frac{1}{\frac{a_k}{a_{k-1}} + \frac{a_{k-1}}{a_k} - 2}}}.$$

Let $b_k = \frac{a_{k+1}}{a_k}$. Then, $b_0 = 1 + \frac{1}{\sqrt{3}}$ by $\tau_K = \varphi_K = 1$, and

$$b_k = 1 + \frac{1}{\sqrt{1 + \frac{1}{\left(\sqrt{b_{k-1}} - \sqrt{\frac{1}{b_{k-1}}}\right)^2}}} \quad \Longleftrightarrow \quad \frac{1}{(b_k - 1)^2} = 1 + \frac{1}{(b_{k-1} - 1)} + \frac{1}{(b_{k-1} - 1)^2}.$$

Let $c_k = \frac{1}{b_k - 1}$. Then

$$c_k^2 = c_{k-1}^2 + c_{k-1} + 1$$

where $c_0 = \sqrt{3}$. Also, by definition,

$$a_{k+1} = b_k b_{k-1} \ldots b_0 = \left(1 + \frac{1}{c_k}\right)\left(1 + \frac{1}{c_{k-1}}\right) \ldots \left(1 + \frac{1}{c_0}\right).$$

Using $c_k^2 = c_{k-1}^2 + c_{k-1} + 1 \iff \frac{c_k^2 - 1}{c_{k-1}^2} = 1 + \frac{1}{c_{k-1}}$, we have

$$\left(\frac{c_k + 1}{c_k}\right)\left(\frac{c_{k-1} + 1}{c_{k-1}}\right) \ldots \left(\frac{c_0 + 1}{c_0}\right) = \frac{c_{k+1}^2 - 1}{c_k^2} \frac{c_k^2 - 1}{c_{k-1}^2} \ldots \frac{c_1^2 - 1}{c_0^2}$$

$$= \left(\frac{c_{k+1} + 1}{c_k}\right)\left(\frac{c_k + 1}{c_{k-1}}\right) \ldots \left(\frac{c_1 + 1}{c_0}\right)\left(\frac{c_{k+1} - 1}{c_k}\right)\left(\frac{c_k - 1}{c_{k-1}}\right) \ldots \left(\frac{c_1 - 1}{c_0}\right).$$

And after reduction of fraction, we get

$$\left(\frac{c_{k+1} + 1}{c_0 + 1}\right)\left(\frac{c_{k+1} - 1}{c_k}\right)\left(\frac{c_k - 1}{c_{k-1}}\right) \ldots \left(\frac{c_1 - 1}{c_0}\right) = 1$$

$$\iff c_{k+1}^2 - 1 = (c_0^2 - 1)\left(\frac{c_k}{c_k - 1}\right)\left(\frac{c_{k-1}}{c_{k-1} - 1}\right) \ldots \left(\frac{c_0}{c_0 - 1}\right).$$

$\frac{c_{k-2} + 1}{c_k} \geq \frac{c_{k-2}}{c_{k-1}} \iff c_k \geq c_{k-2} + 1$ since $c_k^2 = (c_{k-1} + \frac{1}{2})^2 + \frac{3}{4}$ implies $c_k \geq c_{k-1} + \frac{1}{2}$. Therefore,

$$\left(\frac{c_k + 1}{c_k}\right)\left(\frac{c_{k-1} + 1}{c_{k-1}}\right) \ldots \left(\frac{c_0 + 1}{c_0}\right) \geq \frac{(c_k + 1)(c_{k-1} + 1)}{c_k c_{k-1}} \frac{(c_1 - 1)(c_0 - 1)}{c_1 c_0 (c_0^2 - 1)}(c_{k+1}^2 - 1).$$

Furthermore, we can show $c_k \geq \frac{k+3}{2}$ by induction. ($c_0 = \sqrt{3} \geq \frac{3}{2}$ and if $c_k \geq \frac{k+3}{2}$, $c_{k+1} \geq c_k + \frac{1}{2} \geq \frac{k+4}{2}$.)

Finally

$$a_{k+1} \geq \frac{(c_1 - 1)(c_0 - 1)}{c_1 c_0 (c_0^2 - 1)}(c_{k+1}^2 - 1) \geq \frac{1}{33}(k + 3)^2.$$

For $k = K - 1$, $\frac{1}{a_K} = \tau_0$ and we get wanted result.

$\square$

# F  Omitted proofs of Section 3

We present (**G-FGM-G**) for the smooth convex setup

$$x_k = \frac{\varphi_{k+1}}{\varphi_k} x_{k-1}^+ + \left(1 - \frac{\varphi_{k+1}}{\varphi_k}\right) z_k$$

$$z_{k+1} = z_k - \frac{\tau_k \varphi_k - \tau_{k+1} \varphi_{k+1}}{L} \nabla f(x_k)$$

for $k = 0, 1, \ldots, K$ where $z_0 = x_0$, $L$ is smoothness constant of $f$, and the nonnegative sequence $\{\varphi_k\}_{k=0}^{K+1}$ and the nondecreasing nonnegative sequence $\{\tau_k\}_{k=0}^{K}$ satisfy $\varphi_{K+1} = 0$, $\varphi_K = \tau_K = 1$, and

$$\tau_k \varphi_k - \tau_{k+1} \varphi_{k+1} = \varphi_{k+1}(\tau_{k+1} - \tau_k) + 1, \qquad (\tau_k \varphi_k - \tau_{k+1} \varphi_{k+1})(\tau_{k+1} - \tau_k) \le \tau_{k+1}.$$

for $k = 0, 1, \ldots, K - 1$.

Note that G-FISTA-G had the parameter requirement $\le \frac{\tau_{k+1}}{2}$ while G-FGM-G (and later G-Güler G) has $\le \tau_{k+1}$. The parameter requirements are otherwise identical.

**Theorem 7.** *Consider* (P) *with* $g = 0$. *G-FGM-G's* $x_K$ *exhibits the rate*

$$\|\nabla f(x_K)\|^2 \le 2L\tau_0(f(x_0) - f_\star).$$

*Proof.* For $k = 0, 1, \ldots, K$, define

$$U_k = \tau_k \left( \frac{1}{2L} \|\nabla f(x_K)\|^2 + \frac{1}{2L} \|\nabla f(x_k)\|^2 + f(x_k) - f(x_K) - \langle \nabla f(x_k), x_k - x_{k-1}^+ \rangle \right)$$
$$+ \frac{L}{\varphi_k} \langle z_k - x_{k-1}^+, z_k - x_K^+ \rangle.$$

(Note that $z_K = x_K$.) By plugging in the definitions and performing direct calculations, we get

$$U_K = \frac{1}{L} \|\nabla f(x_k)\|^2 \qquad \text{and} \qquad U_0 = \tau_0 \left( \frac{1}{2L} \|\nabla f(x_K)\|^2 + \frac{1}{2L} \|\nabla f(x_0)\|^2 + f(x_0) - f(x_K) \right).$$

We can show that $\{U_k\}_{k=0}^K$ is nonincreasing. Using $\frac{1}{2L} \|\nabla f(x_K)\|^2 \le f(x_K) - f(x_K^+) \le f(x_K) - f(x_\star)$ and $\frac{1}{2L} \|\nabla f(x_0)\|^2 \le f(x_0) - f(x_0^+) \le f(x_0) - f_\star$, which follows from $L$-smoothness, we conclude the rate with

$$\frac{1}{L} \|\nabla f(x_K)\|^2 = U_K \le U_0 \le 2\tau_0 (f(x_0) - f_\star).$$

Now we complete the proof by showing that $\{U_k\}_{k=0}^K$ is nonincreasing. For $k = 0, 1, \ldots K - 1$, we have

$$0 \ge \tau_{k+1} \left( f(x_{k+1}) - f(x_k) - \langle \nabla f(x_{k+1}), x_{k+1} - x_k \rangle + \frac{1}{2L} \|\nabla f(x_{k+1}) - \nabla f(x_k)\|^2 \right)$$
$$+ (\tau_{k+1} - \tau_k) \left( f(x_k) - f(x_K) - \langle \nabla f(x_k), x_k - x_K \rangle + \frac{1}{2L} \|\nabla f(x_k) - \nabla f(x_K)\|^2 \right)$$
$$= \tau_{k+1} \left( \frac{1}{2L} \|\nabla f(x_K)\|^2 + \frac{1}{2L} \|\nabla f(x_{k+1})\|^2 + f(x_{k+1}) - f(x_K) - \langle \nabla f(x_{k+1}), x_{k+1} - x_k^+ \rangle \right)$$
$$- \tau_k \left( \frac{1}{2L} \|\nabla f(x_K)\|^2 + \frac{1}{2L} \|\nabla f(x_k)\|^2 + f(x_k) - f(x_K) - \langle \nabla f(x_k), x_k - x_{k-1}^+ \rangle \right)$$
$$\underbrace{- \langle \nabla f(x_k), \tau_{k+1} x_k^+ - \tau_k x_{k-1}^+ - (\tau_{k+1} - \tau_k) x_K^+ \rangle}_{:=T},$$

where the inequality follows from the cocoercivity inequalities. Finally, we analyze $T$ with the following geometric argument. Let $t \in \mathbb{R}^n$ be the projection of $x_K^+$ onto the plane of iteration. Then,

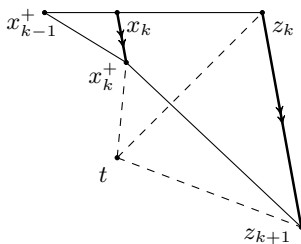

Figure 8: Plane of iteration of G-FGM-G

$$
\frac{1}{L}T \overset{(i)}{=} \left\langle \overrightarrow{x_k x_k^+}, (\tau_{k+1} - \tau_k)t\overrightarrow{x_k^+} + \tau_k \overrightarrow{x_{k-1}^+ x_k^+} \right\rangle
$$

$$
\overset{(ii)}{=} \left\langle \overrightarrow{x_k x_k^+}, (\tau_{k+1} - \tau_k)\left( \overrightarrow{tz_{k+1}} - \overrightarrow{z_k z_{k+1}} - \overrightarrow{x_k z_k} + \overrightarrow{x_k x_k^+} \right) + \tau_k \left( \overrightarrow{x_{k-1}^+ x_k} + \overrightarrow{x_k x_k^+} \right) \right\rangle
$$

$$
\overset{(iii)}{=} \left\langle \overrightarrow{x_k x_k^+}, (\tau_{k+1} - \tau_k)\overrightarrow{tz_{k+1}} - (\tau_{k+1} - \tau_k)(\tau_k\varphi_k - \tau_{k+1}\varphi_{k+1} - 1)\overrightarrow{x_k x_k^+} \right.
$$
$$
\left. + \tau_k \overrightarrow{x_k x_k^+} - (\tau_{k+1} - \tau_k)\overrightarrow{x_k z_k} + \tau_k \left( \frac{\varphi_k}{\varphi_{k+1}} - 1 \right) \overrightarrow{x_k z_k} \right\rangle
$$

$$
\overset{(iv)}{\geq} \left\langle \overrightarrow{x_k x_k^+}, (\tau_{k+1} - \tau_k)\overrightarrow{tz_{k+1}} + \frac{\tau_k\varphi_k - \tau_{k+1}\varphi_{k+1}}{\varphi_{k+1}}\overrightarrow{x_k z_k} \right\rangle
$$

$$
\overset{(v)}{=} \frac{1}{\varphi_{k+1}} \left\langle \overrightarrow{x_k^+ z_{k+1}} - \overrightarrow{x_k z_k}, \overrightarrow{tz_{k+1}} \right\rangle + \frac{1}{\varphi_{k+1}} \left\langle \overrightarrow{tz_{k+1}} - \overrightarrow{tz_k}, \overrightarrow{x_k z_k} \right\rangle
$$

$$
\overset{(vi)}{=} \frac{1}{\varphi_{k+1}} \left\langle z_{k+1} - x_k^+, z_{k+1} - x_K^+ \right\rangle - \frac{1}{\varphi_k} \left\langle z_k - x_{k-1}^+, z_k - x_K^+ \right\rangle,
$$

where (i) follows from the definition of $t$ and the fact that we can replace $x_K^+$ with $t$, the projection of $x_K$ onto the plane of iteration, without affecting the inner products, (ii) from vector addition, (iii) from the fact that $\overrightarrow{x_k x_k^+}$ and $\overrightarrow{z_k z_{k+1}}$ are parallel and their lengths satisfy $(\tau_k\varphi_k - \tau_{k+1}\varphi_{k+1})\overrightarrow{x_k x_k^+} = \overrightarrow{z_k z_{k+1}}$ and $\overrightarrow{x_{k-1}^+ x_k}$ and $\overrightarrow{x_k z_k}$ are parallel and their lengths satisfy $\left( \frac{\varphi_k}{\varphi_{k+1}} - 1 \right) \overrightarrow{x_k z_k} = \overrightarrow{x_{k-1}^+ x_k}$, (iv) from vector addition and

$$
\tau_{k+1} - (\tau_k\varphi_k - \tau_{k+1}\varphi_{k+1})(\tau_{k+1} - \tau_k) \geq 0, \tag{9}
$$

(v) from distributing the product and substituting $\overrightarrow{x_k x_k^+} = (\tau_k\varphi_k - \tau_{k+1}\varphi_{k+1} - 1)^{-1} \left( \overrightarrow{x_k^+ z_{k+1}} - \overrightarrow{x_k z_k} \right) = (\varphi_{k+1}(\tau_{k+1} - \tau_k))^{-1} \left( \overrightarrow{x_k^+ z_{k+1}} - \overrightarrow{x_k z_k} \right)$ into the first term and $\overrightarrow{x_k x_k^+} = (\tau_k\varphi_k - \tau_{k+1}\varphi_{k+1})^{-1} \overrightarrow{z_k z_{k+1}} = (\tau_k\varphi_k - \tau_{k+1}\varphi_{k+1})^{-1} \left( \overrightarrow{tz_{k+1}} - \overrightarrow{tz_k} \right)$ into the second term, and (vi) from cancelling out the cross terms, using $\varphi_k^{-1}\overrightarrow{x_{k-1}^+ z_k} = \varphi_{k+1}^{-1}\overrightarrow{x_k z_k}$, and by replacing $t$ with $x_K^+$ in the inner products. In (v), we also used

$$
\tau_k\varphi_k - \tau_{k+1}\varphi_{k+1} = \varphi_{k+1}(\tau_{k+1} - \tau_k) + 1. \tag{10}
$$

Thus we conclude $U_{k+1} \leq U_k$ for $k = 0, 1, \dots, K - 1$. $\qquad\square$

**Theorem 8.** *Consider* (P) *with* $g = 0$. *FGM-G's* $x_K$ *exhibits the rate*

$$
\|\nabla f(x_K)\|^2 \leq \frac{66L}{(K+2)^2} (f(x_0) - f_\star).
$$

*Proof.* This follows from plugging FGM-G's $\varphi_k$ and $\frac{2\varphi_{k-1}}{(\varphi_{k-1}-\varphi_k)^2}$ into Theorem 7's $\varphi_k$ and $\tau_k$. We can check condition (9), condition (10), and

$$\varphi_k \tau_k - \tau_{k+1}\varphi_{k+1} = \varphi_{k+1}(\tau_{k+1} - \tau_k) + 1 = \frac{\varphi_k}{\varphi_k - \varphi_{k+1}}.$$

Then using Lemma 2, we get the iteration of the form of FGM-G. $\qquad\square$

We present (**G-Güler-G**) for the proximal-point setup:

$$x_k = \frac{\varphi_{k+1}}{\varphi_k}x_{k-1}^{\circ} + \left(1 - \frac{\varphi_{k+1}}{\varphi_k}\right)z_k$$

$$z_{k+1} = z_k - (\tau_k \varphi_k - \tau_{k+1}\varphi_{k+1})\,\lambda\tilde{\nabla}_{1/\lambda}g(x_k)$$

for $k = 0, 1, \ldots, K$ where $z_0 = x_0$ and the nonnegative sequence $\{\varphi_k\}_{k=0}^{K+1}$ and the nondecreasing nonnegative sequence $\{\tau_k\}_{k=0}^{K}$ satisfy $\varphi_{K+1} = 0$, $\varphi_K = \tau_K = 1$, and

$$\tau_k \varphi_k - \tau_{k+1}\varphi_{k+1} = \varphi_{k+1}(\tau_{k+1} - \tau_k) + 1, \qquad (\tau_k \varphi_k - \tau_{k+1}\varphi_{k+1})(\tau_{k+1} - \tau_k) \le \tau_{k+1}.$$

for $k = 0, 1, \ldots, K-1$.

**Theorem 9.** *Consdier* (P) *with* $f = 0$. *G-Güler-G's* $x_K$ *exhibits the rate*

$$\|\tilde{\nabla}_{1/\lambda}g(x_K)\|^2 \le \frac{\tau_0}{\lambda}\left(g(x_0) - g_\star\right)$$

*Proof.* For $k = 0, 1, \ldots, K$, define

$$U_k = \tau_k\left(\lambda\left\|\tilde{\nabla}_{1/\lambda}g(x_k)\right\|^2 + g(x_k^{\circ}) - g(x_K^{\circ}) - \tilde{\nabla}_{1/\lambda}g(x_k)\cdot(x_k - x_{k-1}^{\circ})\right) + \frac{1}{\lambda\varphi_k}\left\langle z_k - x_{k-1}^{\circ}, z_k - x_K^{\circ}\right\rangle.$$

(Note that $z_K = x_K$.) By plugging in the definitions and performing direct calculations, we get

$$U_K = \lambda\left\|\tilde{\nabla}_{1/\lambda}g(x_K)\right\|^2 \qquad\text{and}\qquad U_0 = \tau_0\left(\lambda\left\|\tilde{\nabla}_{1/\lambda}g(x_0)\right\|^2 + g(x_0^{\circ}) - g(x_K^{\circ})\right).$$

We can show that $\{U_k\}_{k=0}^{K}$ is nonincreasing. Using $\lambda\left\|\tilde{\nabla}_{1/\lambda}g(x_0)\right\|^2 \le g(x_0) - g(x_0^{\circ})$, we conclude the rate with

$$\lambda\left\|\tilde{\nabla}_{1/\lambda}g(x_K)\right\|^2 = U_K \le U_0 \le \tau_0\left(g(x_0) - g(x_K^{\circ})\right) \le \tau_0\left(g(x_0) - g_\star\right).$$

Now we complete the proof by showing that $\{U_k\}_{k=0}^{K}$ is nonincreasing. For $k = 0, 1, \ldots, K-1$, we have

$$0 \ge \tau_{k+1}\left(g(x_{k+1}^{\circ}) - g(x_k^{\circ}) - \left\langle\tilde{\nabla}_{1/\lambda}g(x_{k+1}), x_{k+1} - x_k^{\circ}\right\rangle + \lambda\left\|\tilde{\nabla}_{1/\lambda}g(x_{k+1})\right\|^2\right)$$

$$+ (\tau_{k+1} - \tau_k)\left(g(x_k^{\circ}) - g(x_K^{\circ}) - \left\langle\tilde{\nabla}_{1/\lambda}g(x_k), x_k - x_K^{\circ}\right\rangle + \lambda\left\|\tilde{\nabla}_{1/\lambda}g(x_k)\right\|^2\right)$$

$$= \tau_{k+1}\left(\lambda\left\|\tilde{\nabla}_{1/\lambda}g(x_{k+1})\right\|^2 + g(x_{k+1}^{\circ}) - g(x_K^{\circ}) - \tilde{\nabla}_{1/\lambda}g(x_{k+1})\cdot(x_{k+1} - x_k^{\circ})\right)$$

$$- \tau_k\left(\lambda\left\|\tilde{\nabla}_{1/\lambda}g(x_k)\right\|^2 + g(x_k^{\circ}) - g(x_K^{\circ}) - \tilde{\nabla}_{1/\lambda}g(x_k)\cdot(x_k - x_{k-1}^{\circ})\right)$$

$$\underbrace{- \left\langle\tilde{\nabla}_{1/\lambda}g(x_k^{\circ}), \tau_{k+1}x_k^{\circ} - \tau_k x_{k-1}^{\circ} - (\tau_{k+1} - \tau_k)x_K^{\circ}\right\rangle}_{:=T},$$

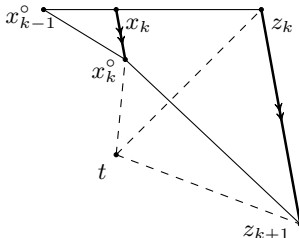

Figure 9: Plane of iteration of G-Güler-G

where the inequality follows from the convexity inequalities. Finally, we analyze $T$ with the following geometric argument. Let $t \in \mathbb{R}^n$ be the projection of $x_K^\circ$ onto the plane of iteration. Then

$$
\begin{aligned}
\frac{1}{L}T \overset{(i)}{=}& \left\langle \overrightarrow{x_k x_k^\lozenge}, (\tau_{k+1} - \tau_k)\overrightarrow{t x_k^\lozenge} + \tau_k \overrightarrow{x_{k-1}^\circ x_k^\lozenge} \right\rangle \\
\overset{(ii)}{=}& \left\langle \overrightarrow{x_k x_k^\lozenge}, (\tau_{k+1} - \tau_k)\left( \overrightarrow{t z_{k+1}} - \overrightarrow{z_k z_{k+1}} - \overrightarrow{x_k z_k} + \overrightarrow{x_k x_k^\lozenge} \right) + \tau_k \left( \overrightarrow{x_{k-1}^\circ x_k} + \overrightarrow{x_k x_k^\lozenge} \right) \right\rangle \\
\overset{(iii)}{=}& \left\langle \overrightarrow{x_k x_k^\lozenge}, (\tau_{k+1} - \tau_k)\overrightarrow{t z_{k+1}} - (\tau_{k+1} - \tau_k)(\tau_k \varphi_k - \tau_{k+1}\varphi_{k+1} - 1)\overrightarrow{x_k x_k^\lozenge} \right. \\
& \left. + \tau_k \overrightarrow{x_k x_k^\lozenge} - (\tau_{k+1} - \tau_k)\overrightarrow{x_k z_k} + \tau_k \left( \frac{\varphi_k}{\varphi_{k+1}} - 1 \right)\overrightarrow{x_k z_k} \right\rangle \\
\overset{(iv)}{\geq}& \left\langle \overrightarrow{x_k x_k^\lozenge}, (\tau_{k+1} - \tau_k)\overrightarrow{t z_{k+1}} + \frac{\tau_k \varphi_k - \tau_{k+1}\varphi_{k+1}}{\varphi_{k+1}}\overrightarrow{x_k z_k} \right\rangle \\
\overset{(v)}{=}& \frac{1}{\varphi_{k+1}}\left\langle \overrightarrow{x_k^\circ z_{k+1}} - \overrightarrow{x_k z_k}, \overrightarrow{t z_{k+1}} \right\rangle + \frac{1}{\varphi_{k+1}}\left\langle \overrightarrow{t z_{k+1}} - \overrightarrow{t z_k}, \overrightarrow{x_k z_k} \right\rangle \\
\overset{(vi)}{=}& \frac{1}{\varphi_{k+1}}\left\langle z_{k+1} - x_k^\circ, z_{k+1} - x_K^\circ \right\rangle - \frac{1}{\varphi_k}\left\langle z_k - x_{k-1}^\circ, z_k - x_K^\circ \right\rangle,
\end{aligned}
$$

where (i) follows from the definition of $t$ and the fact that we can replace $x_K^\circ$ with $t$, the projection of $x_K$ onto the plane of iteration, without affecting the inner products, (ii) from vector addition, (iii) from the fact that $\overrightarrow{x_k x_k^\lozenge}$ and $\overrightarrow{z_k z_{k+1}}$ are parallel and their lengths satisfy $(\tau_k \varphi_k - \tau_{k+1}\varphi_{k+1})\overrightarrow{x_k x_k^\lozenge} = \overrightarrow{z_k z_{k+1}}$ and $\overrightarrow{x_{k-1}^\circ x_k}$ and $\overrightarrow{x_k z_k}$ are parallel and their lengths satisfy $\left( \frac{\varphi_k}{\varphi_{k+1}} - 1 \right)\overrightarrow{x_k z_k} = \overrightarrow{x_{k-1}^\circ x_k}$, (iv) from vector addition and

$$
\tau_{k+1} - (\tau_k \varphi_k - \tau_{k+1}\varphi_{k+1})(\tau_{k+1} - \tau_k) \geq 0, \tag{11}
$$

(v) from distributing the product and substituting $\overrightarrow{x_k x_k^\circ} = (\tau_k \varphi_k - \tau_{k+1}\varphi_{k+1} - 1)^{-1}\left( \overrightarrow{x_k^\circ z_{k+1}} - \overrightarrow{x_k z_k} \right) = (\varphi_{k+1}(\tau_{k+1} - \tau_k))^{-1}\left( \overrightarrow{x_k^\circ z_{k+1}} - \overrightarrow{x_k z_k} \right)$ into the first term and $\overrightarrow{x_k x_k^\lozenge} = (\tau_k \varphi_k - \tau_{k+1}\varphi_{k+1})^{-1}\overrightarrow{z_k z_{k+1}} = (\tau_k \varphi_k - \tau_{k+1}\varphi_{k+1})^{-1}\left( \overrightarrow{t z_{k+1}} - \overrightarrow{t z_k} \right)$ into the second term, and (vi) from cancelling out the cross terms, using $\varphi_k^{-1}\overrightarrow{x_{k-1}^\circ z_k} = \varphi_{k+1}^{-1}\overrightarrow{x_k z_k}$, and by replacing $t$ with $x_K^+$ in the inner products.

$$
\tau_k \varphi_k - \tau_{k+1}\varphi_{k+1} = \varphi_{k+1}\left( \tau_{k+1} - \tau_k \right) + 1. \tag{12}
$$

Thus we conclude $U_{k+1} \leq U_k$ for $k = 0, 1, \ldots, K - 1$. □

*Proof of Theorem 2.* The conclusion of Theorem 2 follows from plugging Güler-G's $\varphi_k$ and $\tau_k$ into Theorem 9. If $\tau_k = \theta_k^{-2}$ and $\varphi_k = \theta_k^4$, we can check conditions (11) and (12). Then using Lemma 2, we get the iteration of the form of Güler-G. Combining the $g(x_K^\circ) - g_\star \le \|x_0 - x_\star\|^2/\lambda(K+2)^2$ rate of Güler's second method [37, Theorem 6.1] with rate of Güler-G, we get the rate of Güler+Güler-G:

$$\left\| \tilde{\nabla}_{1/\lambda} g(x_{2K}) \right\|^2 \le \frac{4}{\lambda(K+2)^2}\left(g(x_K^\circ) - g_\star\right) \le \frac{4}{\lambda^2(K+2)^4} \|x_0 - x_\star\|^2 .$$

$\square$

# G   Omitted proofs of Section 4

*Proof of Theorem 3.* In the setup of Theorem 3, define

$$U_k = g(x_{k-1}^\circ) - g_\star - \frac{\mu}{2}\|x_{k-1}^\circ - x_\star\|^2 + \mu\|z_k - x_\star\|^2$$

for $k = 0, 1, \dots$. By plugging in the definitions and performing direct calculations, we get

$$U_0 = g(x_0) - g_\star + \frac{\mu}{2}\|x_0 - x_\star\|^2.$$

We can show that $U_{k+1} \le \left(1 - \frac{1}{\sqrt{q}}\right)^2 U_k$ for $k = -1, 0, \dots$. Using strong convexity, we conclude the rate with

$$\mu\|z_k - x_\star\|^2 \le U_k \le U_0 \le 2(g(x_0) - g_\star).$$

Now we complete the proof by showing that $U_{k+1} \le \left(1 - \frac{1}{\sqrt{q}}\right)^2 U_k$ for $k = -1, 0, \dots$. For $k = 0, 1, \dots$, we have

$$U_{k+1} - \left(1 - \frac{1}{\sqrt{q}}\right)^2 U_k$$

$$= \left(g(x_k^\circ) - g_\star - \frac{\mu}{2}\|x_k^\circ - x_\star\|^2\right) - \left(1 - \frac{1}{\sqrt{q}}\right)^2 \left(g(x_{k-1}^\circ) - g_\star - \frac{\mu}{2}\|x_{k-1}^\circ - x_\star\|^2\right)$$

$$+ \mu\|z_{k+1} - x_\star\|^2 - \mu\left(1 - \frac{1}{\sqrt{q}}\right)^2\|z_k - x_\star\|^2.$$

For calculating the last term of difference, we use $(q-1)x_k - (1 - \sqrt{q})^2 x_{k-1}^\circ = 2(\sqrt{q} - 1)z_k$. Since

$$\mu\|z_{k+1} - x_\star\|^2 = \mu\left\| \frac{1}{\sqrt{q}}\left(x_k - \left(\frac{1}{\mu} + \lambda\right)\tilde{\nabla}_{1/\lambda}g(x_k)\right) + \left(1 - \frac{1}{\sqrt{q}}\right)z_k - x_\star\right\|^2$$

$$= \frac{\mu}{q}\left\|x_k - \left(\frac{1}{\mu} + \lambda\right)\tilde{\nabla}_{1/\lambda}g(x_k) - x_\star\right\|^2 + \mu\left(1 - \frac{1}{\sqrt{q}}\right)^2\|z_k - x_\star\|^2$$

$$+ 2\left(1 - \frac{1}{\sqrt{q}}\right)\frac{\mu}{\sqrt{q}}\left\langle x_k - \left(\frac{1}{\mu} + \lambda\right)\tilde{\nabla}_{1/\lambda}g(x_k) - x_\star, z_k - x_\star\right\rangle,$$

we get

$$\mu\|z_{k+1} - x_\star\|^2 - \mu\left(1 - \frac{1}{\sqrt{q}}\right)^2\|z_k - x_\star\|^2$$

$$= \frac{\mu}{q} \left\| x_k - \left( \frac{1}{\mu} + \lambda \right) \tilde{\nabla}_{1/\lambda} g(x_k) - x_\star \right\|^2$$

$$+ \frac{\mu}{q} \left\langle x_k - \left( \frac{1}{\mu} + \lambda \right) \tilde{\nabla}_{1/\lambda} g(x_k) - x_\star, (q-1)(x_k - x_\star) - (1 - \sqrt{q})^2 (x^\circ_{k-1} - x_\star) \right\rangle$$

$$= \frac{\mu}{q} \left\langle x_k - \left( \frac{1}{\mu} + \lambda \right) \tilde{\nabla}_{1/\lambda} g(x_k) - x_\star, - \left( \frac{1}{\mu} + \lambda \right) \tilde{\nabla}_{1/\lambda} g(x_k) + q(x_k - x_\star) - (1 - \sqrt{q})^2 (x^\circ_{k-1} - x_\star) \right\rangle$$

$$= \frac{\mu}{q} \left\langle x_k - \left( \frac{1}{\mu} + \lambda \right) \tilde{\nabla}_{1/\lambda} g(x_k) - x_\star, q(x^\circ_k - x_\star) - (1 - \sqrt{q})^2 (x^\circ_{k-1} - x_\star) \right\rangle$$

$$= \mu \left\langle x^\circ_k - \frac{1}{\mu} \tilde{\nabla}_{1/\lambda} g(x_k) - x_\star, x^\circ_k - x_\star \right\rangle - \mu \left( 1 - \frac{1}{\sqrt{q}} \right)^2 \left\langle x^\circ_k - \frac{1}{\mu} \tilde{\nabla}_{1/\lambda} g(x_k) - x_\star, x^\circ_{k-1} - x_\star \right\rangle$$

$$= \left( 1 - \left( 1 - \frac{1}{\sqrt{q}} \right)^2 \right) \left\langle \mu x^\circ_k - \tilde{\nabla}_{1/\lambda} g(x_k) - \mu x_\star, x^\circ_k - x_\star \right\rangle + \left( 1 - \frac{1}{\sqrt{q}} \right)^2 \left\langle \mu x^\circ_k - \tilde{\nabla}_{1/\lambda} g(x_k) - \mu x_\star, x^\circ_k - x^\circ_{k-1} \right\rangle.$$

Therefore, we can write difference of $U_{k+1}$ and $\left( 1 - \frac{1}{\sqrt{q}} \right)^2 U_k$ as

$$U_{k+1} - \left( 1 - \frac{1}{\sqrt{q}} \right)^2 U_k$$

$$= \left( g(x^\circ_k) - g_\star - \frac{\mu}{2} \|x^\circ_k - x_\star\|^2 \right) - \left( 1 - \frac{1}{\sqrt{q}} \right)^2 \left( g(x^\circ_{k-1}) - g_\star - \frac{\mu}{2} \|x^\circ_{k-1} - x_\star\|^2 \right)$$

$$+ \left( 1 - \left( 1 - \frac{1}{\sqrt{q}} \right)^2 \right) \left\langle \mu x^\circ_k - \tilde{\nabla}_{1/\lambda} g(x_k) - \mu x_\star, x^\circ_k - x_\star \right\rangle + \left( 1 - \frac{1}{\sqrt{q}} \right)^2 \left\langle \mu x^\circ_k - \tilde{\nabla}_{1/\lambda} g(x_k) - \mu x_\star, x^\circ_k - x^\circ_{k-1} \right\rangle$$

$$= (g(x^\circ_k) - g_\star) - \left( 1 - \frac{1}{\sqrt{q}} \right)^2 \left( g(x^\circ_{k-1}) - g_\star - \frac{\mu}{2} \|x^\circ_{k-1} - x_\star\|^2 + \frac{\mu}{2} \|x^\circ_k - x_\star\|^2 \right)$$

$$- \left( 1 - \left( 1 - \frac{1}{\sqrt{q}} \right)^2 \right) \frac{\mu}{2} \|x^\circ_k - x_\star\|^2 + \left( 1 - \left( 1 - \frac{1}{\sqrt{q}} \right)^2 \right) \left\langle \mu x^\circ_k - \tilde{\nabla}_{1/\lambda} g(x_k) - \mu x_\star, x^\circ_k - x_\star \right\rangle$$

$$+ \left( 1 - \frac{1}{\sqrt{q}} \right)^2 \left\langle \mu x^\circ_k - \tilde{\nabla}_{1/\lambda} g(x_k) - \mu x_\star, x^\circ_k - x^\circ_{k-1} \right\rangle$$

$$= (g(x^\circ_k) - g_\star) - \left( 1 - \frac{1}{\sqrt{q}} \right)^2 (g(x^\circ_{k-1}) - g_\star) + \left( 1 - \frac{1}{\sqrt{q}} \right)^2 \frac{\mu}{2} \left\langle x^\circ_{k-1} - x^\circ_k, x^\circ_{k-1} + x^\circ_k - 2x_\star \right\rangle$$

$$+ \left( 1 - \left( 1 - \frac{1}{\sqrt{q}} \right)^2 \right) \left\langle \frac{\mu}{2} (x^\circ_k - x_\star) - \tilde{\nabla}_{1/\lambda} g(x_k), x^\circ_k - x_\star \right\rangle$$

$$+ \left( 1 - \frac{1}{\sqrt{q}} \right)^2 \left\langle -\mu x^\circ_k + \tilde{\nabla}_{1/\lambda} g(x_k) + \mu x_\star, x^\circ_{k-1} - x^\circ_k \right\rangle$$

$$= (g(x^\circ_k) - g_\star) - \left( 1 - \frac{1}{\sqrt{q}} \right)^2 (g(x^\circ_{k-1}) - g_\star) + \left( 1 - \frac{1}{\sqrt{q}} \right)^2 \left\langle \frac{\mu}{2} (x^\circ_{k-1} - x^\circ_k) + \tilde{\nabla}_{1/\lambda} g(x_k), x^\circ_{k-1} - x^\circ_k \right\rangle$$

$$+ \left( 1 - \left( 1 - \frac{1}{\sqrt{q}} \right)^2 \right) \left\langle \frac{\mu}{2} (x^\circ_k - x_\star) - \tilde{\nabla}_{1/\lambda} g(x_k), x^\circ_k - x_\star \right\rangle$$

$$= -\left(1 - \frac{1}{\sqrt{q}}\right)^2 \left(g(x_{k-1}^\circ) - g(x_k^\circ) - \left\langle \tilde{\nabla}_{1/\lambda} g(x_k), x_{k-1}^\circ - x_k^\circ \right\rangle - \frac{\mu}{2}\|x_{k-1}^\circ - x_k^\circ\|^2\right)$$

$$- \left(1 - \left(1 - \frac{1}{\sqrt{q}}\right)^2\right) \left(g_\star - g(x_k^\circ) - \left\langle \tilde{\nabla}_{1/\lambda} g(x_k), x_\star - x_k^\circ \right\rangle - \frac{\mu}{2}\|x_\star - x_k^\circ\|^2\right)$$

$$\leq 0,$$

where the inequality follows from strong convexity inequalities.

The case $U_1 \leq \left(1 - \frac{1}{\sqrt{q}}\right)^2 U_0$ follows from the same argument with $x_{-1}^\circ = x_0$. Thus $U_{k+1} \leq \left(1 - \frac{1}{\sqrt{q}}\right)^2 U_k$ for $k = -1, 0, \ldots$. $\qquad\square$

Following proof is a close adaptation of the convergence analysis of ITEM [19, Theorem 3].

*Proof of Theorem 4.* In the setup of Theorem 4, define

$$U_k = A_k \left(g(x_{k-1}^\circ) - g_\star - \frac{\mu}{2}\|x_{k-1}^\circ - x_\star\|^2\right) + \left(A_k\mu + \mu + \frac{1}{\lambda}\right)\|z_k - x_\star\|^2$$

for $k = 0, 1, \ldots$. By plugging in the definitions and performing direct calculations, we get

$$U_0 = \left(\mu + \frac{1}{\lambda}\right)\|x_0 - x_\star\|^2.$$

We can show that $\{U_k\}_{k=0}^\infty$ is nonincreasing. Using strong convexity, we conclude the rate with

$$\left(A_k\mu + \mu + \frac{1}{\lambda}\right)\|z_k - x_\star\|^2 \leq U_k \leq U_0 = \left(\mu + \frac{1}{\lambda}\right)\|z_0 - x_\star\|^2.$$

And by

$$A_k = \frac{(1+q)A_{k-1} + 2\left(1 + \sqrt{(1+A_{k-1})(1+qA_{k-1})}\right)}{(1-q)^2} \geq \frac{(1+q)A_{k-1} + 2\sqrt{qA_{k-1}^2}}{(1-q)^2} = \frac{A_{k-1}}{(1-\sqrt{q})^2},$$

we get theorem through direct calculation.

Now we complete the proof by showing that $\{U_k\}_{k=0}^\infty$ is nonincreasing. For $k = 1, 2, \ldots$, we have

$$U_{k+1} - U_k$$

$$= 4\frac{\lambda}{1-q}K_2 P(A_{k+1}, A_k)\left\|(1-q)A_{k+1}\tilde{\nabla}_{1/\lambda}g(x_k) - \frac{\mu}{1+\lambda\mu}A_k(x_{k-1}^\circ - x_\star) + \frac{\mu}{1+\lambda\mu}K_3(z_k - x_\star)\right\|^2$$

$$- \frac{1}{\lambda(1-q)}K_1 P(A_{k+1}, A_k)\|z_k - x_\star\|^2$$

$$+ A_k \left(g(x_k^\circ) - g(x_{k-1}^\circ) + \left\langle \tilde{\nabla}_{1/\lambda}g(x_k), x_{k-1}^\circ - x_k^\circ \right\rangle + \frac{\mu}{2}\|x_{k-1}^\circ - x_k^\circ\|^2\right)$$

$$+ (A_{k+1} - A_k)\left(g(x_k^\circ) - g_\star + \left\langle \tilde{\nabla}_{1/\lambda}g(x_k), x_\star - x_k^\circ \right\rangle + \frac{\mu}{2}\|x_\star - x_k^\circ\|^2\right)$$

$$\leq 0$$

where inequality follows from the strong convexity inequality,

$$K_1 = \frac{q^2}{(1+q)^2 + (1-q)^2 q A_{k+1}}$$

$$K_2 = \frac{(1+q)^2 + (1-q)^2 q A_{k+1}}{(1-q)^2 (1+q+qA_k)^2 A_{k+1}^2}$$

$$K_3 = (1+q)\frac{(1+q)A_k - (1-q)(2+qA_k)A_{k+1}}{(1+q)^2 + (1-q)^2 q A_{k+1}},$$

$P(x,y) = (y - (1-q)x)^2 - 4x(1+qy)$ and $P(A_{k+1}, A_k) = 0$ by condition, and equality follows from direct calculation.

The case $U_1 \leq U_0$ follows from the same argument with $x^\circ_{-1} = x_0$. Thus $U_{k+1} \leq U_k$ for $k = 0, 1, \dots$. $\quad\square$

## H   Other geometric and non-geometric views of acceleration

Geometric descent is an accelerated method designed expressly based on a geometric principle of shrinking balls for the smooth strongly convex setup [13, 16, 42]. Quadratic averaging is equivalent to geometric descent but has an interpretation of averaging quadratic lower bounds [30]. Both methods implicitly induce the collinear structure through steps equivalent to defining $z_{k+1}$ as a convex combination of $z_k$ and $x_k^{++}$. (In fact, our $x_k^{++}$ notation comes from the geometric descent paper [13].) However, this line of work does not establish a rate faster than FGM or its corresponding proximal version, nor does it extend the geometric principle to the non-strongly convex setup.

The method of similar triangles (MST) is an accelerated method [32, 60, 1] with iterates forming similar triangles analogous to our illustration of FGM in Figure 1. One can also interpret acceleration as an approximate proximal point method with alternating upper and lower bounds and obtain the structure of similar triangles as a consequence [1]. The parallel structure we present generalizes the structure of similar triangles; the illustration of OGM and OGM-G in Figure 1 exhibits the parallel structure but not the similar triangles structure. To the best of our knowledge, the parallel structure we present is a geometric structure of acceleration that has not been considered, explicitly or implicitly, in prior works.

Linear coupling [4] interprets acceleration as a unification of gradient descent and mirror descent. The auxiliary iterates of our setup are referred to as the mirror descent iterates in the linear coupling viewpoint. However, the primary motivation of linear coupling is to unify gradient descent, which reduces the function value much when the gradient is large, with mirror descent, which reduces the function value much when the gradient is small. This motivation does not seem to be applicable to the problem setup of minimizing gradient magnitudes, the setup of OGM-G and FISTA-G.

The scaled relative graph (SRG) is another geometric framework for analyzing optimization algorithms; it establishes a correspondence between algebraic operations on nonlinear operators with geometric operations on subsets of the 2D plane [66, 40, 41, 68]. The SRG demonstrated that geometry can serve as a powerful tool for the analysis of optimization algorithms. However, there is no direct connection as the SRG has not been used to analyze *accelerated* optimization algorithms.

## I   Experiment

For scientific reproducibility, we include code for generating the synthetic data of the experiments. We furthermore clarify that since FPGM-m, FISTA, and FISTA+FISTA-G are not anytime algorithms (i.e., since the total iteration count $K$ must be known in advance), the points in the plot of Figure 4 were generated with

a separate iteration. In other words, the plots for ISTA and FISTA were generated each with a single for-loop, while the plots for FPGM-m, FISTA, and FISTA+FISTA-G were generated with nested double for-loops.

```python
import numpy as np

np.random.seed(419)

#l1 norm problem data
m, n, k = 60, 100, 20        # dimensions
lamb = 0.1                   # lasso penalty constant
L = 324
x_true = np.zeros(n)
x_true[:k] = np.random.randn(k)
np.random.shuffle(x_true)
[U,_] = np.linalg.qr(np.random.randn(m,m))
[V,_] = np.linalg.qr(np.random.randn(n,n))
Sigma =  np.zeros((m,n))
np.fill_diagonal(Sigma,np.abs(np.random.randn(m)))
np.fill_diagonal(Sigma[m-3:m,m-3:m],np.sqrt(L))
A = U @ Sigma @ V.T
b = A@x_true + 0.01 * np.random.randn(m)

#nuclear problem data
m, n, k = 60, 20, 20         # dimensions
lamb = 0.1                   # nuclear norm penalty constant
L = 400
n2 = int(n*(n+1)/2)
x_true = np.zeros(n2)
x_true[:k] = np.random.randn(k)
np.random.shuffle(x_true)
[U,_] = np.linalg.qr(np.random.randn(m,m))
[V,_] = np.linalg.qr(np.random.randn(n2,n2))
Sigma =  np.zeros((m,n2))
np.fill_diagonal(Sigma,np.abs(np.random.randn(m)))
np.fill_diagonal(Sigma[m-3:m,m-3:m],np.sqrt(L))
A = U @ Sigma @ V.T
b = A@x_true + 0.01 * np.random.randn(m)
```