# OpenReview forum: "A Geometric Structure of Acceleration and Its Role in Making Gradients Small Fast"
_NeurIPS.cc/2021/Conference — NeurIPS 2021 Poster_

### Official Review · Reviewer_jb5a · 2021-07-08

**Rating:** 6
**Confidence:** 3

**Summary:**

Two main contributions of this paper include (i) proposing a geometric structure that is satisfied by many first-order accelerated methods and (ii) proposing algorithms with the rate $O(1/K^4)$ for the squared gradient norm.

**Limitations And Societal Impact:**

N.A.

**Main Review:**

1. The parallel structure and the collinear structure are quite interesting.
2. The way of determining the extrapolation parameters $\theta_k$ is strange:$ \theta_K =1$, $\theta_k$ is defined depending on $\theta_{k+1}$. Does it mean we must save all the values of the sequence $\theta_k$ in advance before running the algorithm?
3. $U_k$ is a sequence, it is not a Lyapunov function. The authors may want to call it an auxiliary sequence.
4. Although the obtained rate is optimal, it is not clear how to choose K to obtain an $\varepsilon$-optimal solution. This is important because K is chosen in advance for the proposed algorithms.



**Time Spent Reviewing:**

20

---

> ### Author Response · Authors · 2021-08-09
> **Response to Reviewer jb5a**
>
> We thank the reviewer for the thoughtful reviews.
> 1. Thank you. We are glad to hear that the reviewer found our geometric approach interesting.
>
> 2. The reverse recursion of the iteration parameters is indeed unusual. To execute the algorithm one should (a) fix the total iteration count $K$ a priori (b) execute a for-loop to pre-compute $\theta_K,\theta_{K-1},\dots,\theta_0$ and (c) execute a for-loop to perform OGM-G (or FISTA-G).
>
> 3. We referred to $U_k$ as a "Lyapunov function'' because it is the key nonincreasing quantity that we use to establish the convergence result. However, we are not quite familiar with the formal definition of what constitutes a Lyapunov function. (None of the authors are control theorists.) Can the reviewer clarify why it is that $U_k$ is not a Lyapunov function?
>
> 4. The reviewer makes a very good point. When we translate the bounds, obtaining an iterate $x_K$ with $\\| \widetilde{\nabla_L} F(x_K)\\|\^2\le \varepsilon$ requires the total iteration count $K$ to satisfy
> $$K\ge \left({\frac{4L(f(x_0)-f_{\star})}{\varepsilon}}\right)^{\frac{1}{2}}$$
> for OGM-G
> $$K\ge \left(\frac{66L(F(x_0)-F_{\star})}{\varepsilon}\right)^{\frac{1}{2}}$$
> for FISTA-G, and
> $$K\ge 2\left(\frac{132L^2\\|x_0-x_{\star}\\|^2}{\varepsilon}\right)^{\frac{1}{4}}$$ (K must be positive even interger)  for FISTA+FISTA-G.
> We will add these translated bounds in the revision.

---

### Official Review · Reviewer_wtVu · 2021-07-13

**Rating:** 2
**Confidence:** 5

**Summary:**

This paper tries to propose a new algorithm to obtain faster rate O(1/k^4) for convex objective function. The bound is OK on the surface, actually does not make sense.

**Limitations And Societal Impact:**

I hope the authors could consider a real algorithms in practice, not only give a bound and a superficial proof.

**Main Review:**

The main algorithm (FISTA-G) between the line 114 and the line 115 does not work practically. The key coefficient psi_k needs the afterword coefficient psi_{k+1} and psi_{k+2}. Moreover, for every specific iteration number K, it needs a special coefficients, psi_{0}, ..., psi_{K+2}. In other words, for different K, these settings are not an algorithm but a series of algorithms, and every algorithm only efficient for a specific K. Essentially, it is not an algorithm and then obtains a bound O(1/K^4) for it. I believe this paper only hopes to use some tricks to obtain a desired bound and does not design an efficient algorithm.

**Time Spent Reviewing:**

40

---

> ### Author Response · Authors · 2021-08-09
> **Response to Reviewer wtVu**
>
> We point out that the reviewer's claim that our algorithm "does not make sense" is wholly mistaken.
> Requiring the total iteration count $K$ of the method to be determined a priori (using $K$ in the determination of the stepsizes) is a common property among stochastic gradient methods with convergence rates. For example, see [6, Algorithms 1-2] or [7, Theorem 3.1].
> Given that the total iteration count $K$ is chosen before the start of the algorithm, it is unclear why only providing a bound for the final iterate is considered disqualifying by the reviewer. Since its discovery in 2018, the acceleration mechanism of OGM-G (which shares a similar algorithmic structure with FISTA-G) has been an active object of study in the field [1-5].
>
> To practically implement our algorithm FISTA-G (or OGM-G) in practice, we first compute the scalar coefficients $\psi_K,\psi_{K-1},\dots,\psi_0$ in one for-loop, and then the actual gradient method is performed in a second for-loop. (The for-loops are not nested.) The cost of computing $\psi_K,\psi_{K-1},\dots,\psi_0$ is small, so this step of pre-computing the coefficients is at most a minor inconvenience and does not significantly add to the total computational cost. We recognize that this issue was not made clear in our writing, so we will further clarify this point in our revision.
>
>
> [1] Y. Nesterov, A. Gasnikov, S. Guminov, and P. Dvurechensky. Primal–dual accelerated gradient methods with small-dimensional relaxation oracle. Optimization Methods and Software, pages 1–38, 2020.
>
> [2] D. Kim and J. A. Fessler. Generalizing the optimized gradient method for smooth convex minimization. SIAM Journal on Optimization, 28(2):1920–1950, 2018.
>
> [3] D. Kim and J. A. Fessler. Optimizing the efficiency of first-order methods for decreasing the gradient of smooth convex functions. Journal of Optimization Theory and Applications, 188(1):192–219, 2021.
>
> [4] J. Diakonikolas and P. Wang. Potential function-based framework for making the gradients small in convex and min-max optimization. arXiv preprint arXiv:2101.12101, 2021.
>
> [5] K. Zhou, L. Tian, A. Man-Cho So, and J. Cheng. Practical Schemes for Finding Near-Stationary Points of Convex Finite-Sums. arXiv preprint arXiv:2105.12062, 2021.
>
> [6] Z. Allen-Zhu. How to make the gradients small stochastically: Even faster convex and nonconvex SGD. NeurIPS, 2018.
>
> [7] Damek Davis and Dmitriy Drusvyatskiy. Stochastic Model-Based Minimization of Weakly Convex Functions. SIAM Journal on Optimization, 29(1): 207–239, 2019.

---

### Official Review · Reviewer_TTHa · 2021-07-16

**Rating:** 7
**Confidence:** 3

**Summary:**

The paper discovers parallel and co-linear structures in known accelerated first order methods and claims that these are important ingredients to achieve acceleration. The introduction of parallel structure in an optimization algorithm yields a standard way of designing a discrete Lyapunov function to analyze its convergence. A new Lyapunov analysis for OGM-G is obtained, but also a new method FISTA-G, which, combined with FISTA, yields to an improvement in the state of the art regarding convergence of gradient in the proximal gradient setting. An experiment in compressed sensing is presented as well.

**Limitations And Societal Impact:**

The authors discuss limitations of the work and there is no negative societal impact.

**Main Review:**

The mechanism behind acceleration is a vastly studied problem in optimization community and I think that any contribution to that is of interest for neurips. The paper finds common characteristics of popular accelerated algorithms. More specifically, these algorithms can be written as a primary and secondary sequence of iterates that satisfy parallel properties in the convex case and co-linear properties in the strongly convex case. What I like here is that the co-linear structure "converges" in some sense to the parallel one when the strong convexity constant goes to $0$.

The authors believe that these can serve as unified characteristics of all accelerated methods and a first remarkable result is that changing slightly the auxiliary sequence of iterates in OGM-G, can yield to a novel Lyapunov analysis of convergence. However, the OGM-G presented in the paper is the same as the original one? The auxiliary iterates participate also in the definition of the primary ones...

After a modification in FISTA to make the auxiliary iterates parallel, the new method FISTA-G combined with FISTA yields $O(1/K^4)$ iteration complexity for gradient convergence. This is better than the sate of the art which is $O(1/K^3)$. Excuse my ignorance here but why FISTA+FISTA-G yields better result than FISTA-G running for all $2K$ iterates? Can you give some intuition? Also, what is the difficulty to establish iteration complexity of the new method for the suboptimality of function values?

Similar things happening for methods for accelerated strongly convex optimization where the auxiliary iterates are endowed with the co-linear structure. In 228 it is claimed that the new methods improve over the original ones, can you explain exactly how? It is not obvious to me.

The paper is in general well-written and explains in detail subtle differences among a very large literature in accelerated methods. However, the figures are not very enlightening to me and I find a bit weird the use of arrows above vectors.

Limitations of the work are discussed at the end, noting that the presented algorithms are not anytime algorithms, but this is also a limitation of the original methods that upon them the paper improves.

Overall, I find the content of the paper a bit out of the ordinary, but since the presented observations can yield to better convergence result even in the special case of prox-grad, I think that this is a good contribution and vote for acceptance. However, I was not able to check the math in detail in the given time which means that my evaluation is not confident. The only concern that i have is that maybe this paper is not such a good fit for neurips due to its highly technical nature (40 pages, mostly math). Probably an optimization journal would be a better fit for proper reviewing.







**Time Spent Reviewing:**

12

---

> ### Author Response · Authors · 2021-08-09
> **Response to Reviewer TTHa**
>
> We thank the reviewer for the highly detailed and insightful feedback. We are happy to hear that the reviewer found our contribution interesting.
>
> 1.	The OGM-G we analyze is indeed identical to the OGM-G presented by Kim and Fessler [3]. OGM-G as stated by Kim and Fessler is
> $$
> x_{k+1}=x_{k}^{+}+ \frac{(\theta_k-1)(2\theta_{k+1}-1)}{\theta_{k}(2\theta_{k}-1)}(x_{k}^{+}-x_{k-1}^{+})+\frac{2\theta_{k+1}-1}{2\theta_{k}-1}(x_{k}^{+}-x_k)
> $$
> for $k=0,1,\dots$.
> Since $\theta_{k+1}^2=\theta_k(\theta_k-1)$ for $k=1,\dots,K$, we have $$\frac{(\theta_k-1)(2\theta_{k+1}-1)}{\theta_{k}(2\theta_{k}-1)}=\frac{\frac{\theta_{k}-1}{2\theta_{k}-1}}{\frac{\theta_{k}}{2\theta_{k+1}-1}}=\frac{\frac{(\theta_{k}-1)(\theta_{k}-1)}{2\theta_{k}-1}}{\frac{\theta_{k}(\theta_{k}-1)}{2\theta_{k+1}-1}}= \frac{\frac{\theta_{k}^2}{2\theta_{k}-1}-1}{\frac{\theta_{k+1}^2}{2\theta_{k+1}-1}},$$ and we can reformulate OGM-G as
> $$
> x_{k+1} =x_k^{+}+\frac{\frac{\theta_{k}^2}{2\theta_{k}-1}-1}{\frac{\theta_{k+1}^2}{2\theta_{k+1}-1}}(x_{k}^+-x_{k-1}^+)+\frac{2\theta_{k+1}-1}{2\theta_{k}-1}(x_{k}^+-x_k)
> $$
> for $k=1,2,…, K$.
> Then we apply Lemma 1 of the appendix with $a_k= \frac{\theta_k^2}{2\theta_k –1}$ and $b_{k+1}= \frac{2\theta_{k+1} –1}{2\theta_k –1}$ to obtain $\varphi_k= \frac{1}{\theta_{k+1}^4}$, $\varphi_{k-1}= \frac{1}{\theta_{k}^4}$, and $z_{k+1} = z_{k}-\frac{\theta_{k}}{L}\nabla f(x_k) $.
> For $k=0$, since $2\theta_{1}^2=\theta_0(\theta_0-1)$, we have  $\frac{(\theta_0-1)(2\theta_{1}-1)}{\theta_{0}(2\theta_{0}-1)}=\frac{\frac{\theta_{0}-1}{2\theta_{0}-1}}{\frac{\theta_{0}}{2\theta_{1}-1}}=\frac{\frac{(\theta_{0}-1)(\theta_{0}-1)}{2(2\theta_{k}-1)}}{\frac{\theta_{0}(\theta_{0}-1)}{2(2\theta_{1}-1)}}= \frac{\frac{\theta_{0}^2+2\theta_0-1}{2(2\theta_{0}-1)}-1}{\frac{\theta_{1}^2}{2\theta_{1}-1}}$. Again, we apply Lemma 1 of the appendix with $a_0=\frac{\theta_0^2+2\theta_0-1}{2(2\theta_0 –1)}$ and $b_{1}= \frac{2\theta_{1} –1}{2\theta_0 –1}$ to obtain $z_1=z_0-\frac{1+\theta_0}{2L}\nabla f(x_0)$.
> Finally, the equivalent form of OGM with the auxiliary iterates is
> $$
> x_{k} = \frac{\theta_{k+1}^4}{\theta_{k}^4}x_{k-1}^{+}+ \left(1- \frac{\theta_{k+1}^4}{\theta_{k}^4}\right)z_{k}
> $$
> $$z_{k+1} = z_{k}-\frac{\theta_{k}}{L}\nabla f(x_k)
> $$
> for $k = 1,2, \dots, K$, where  $z_1=z_0-\frac{1+\theta_0}{2L}\nabla f(x_0)$ and $z_0=x_0$.
> Indeed, the auxiliary $z$-iterates now participate in the algorithm.
>
>
> 2. I think it is helpful to consider optimization algorithms not just as a procedure to find an approximate solution, but rather a procedure providing a terminal guarantee on the accuracy of its approximate solution *given a starting point* with certain guarantees (initial condition). The initial condition to terminal guarantee of FISTA is $\left(\\|x_0-x_\star\\|^2 \mapsto F(x_K)-F(x_\star)\right)$ and for FISTA-G it is $\left(F(x_0)-F(x_\star) \mapsto \\| \widetilde{\nabla_L} F(x_K)\\|\^2\right)$. Therefore, the concatenation FISTA+FISTA-G provides the initial condition to terminal guarantee$\left(\\|x_0-x_\star\\|^2 \mapsto\\| \widetilde{\nabla_L} F(x_{2K})\\|\^2\right)$.
> As to why it is that FISTA-G by itself is slower than FISTA-G, my intuition is that FISTA-G's acceleration mechanism (and OGM-G's acceleration mechanism) hinges on having a small function value at the beginning or at some point of the algorithm. However, FISTA-G by itself does not achieve a very small function value.
> This discussion does raise the question of whether a direct (without concatenation) method with an initial condition to terminal guarantee of ($\\|x_0-x_\star\\|^2 \mapsto\\| \widetilde{\nabla_L} F(x_K)\\|\^2$) with rate $\mathcal{O}(1/K^4)$ can have a better constant. This is an interesting direction of future work.
>
>
> 3. We believe it should be possible to establish a function-value rate for FISTA-G, but we did not pursue this analysis as we do not expect the rate to be good. (It will likely be slower than FISTA, and it will likely not be $\mathcal{O}(1/k^2)$.) In the smooth convex setup, there is some numerical evidence (cf. Table 2 of [3]) indicating that OGM-G is not effective at reducing the function value (but doing so is not the goal of the algorithm).
>
>
> 4. Yes, we can clarify. Proximal-TMM is faster than the variant of the proximal point methods "inexact accelerated proximal point method" [9, Corollary 4.4, v2 version on arXiv] and the "accelerated hybrid proximal extragradient method" [8, Corollary 5.8]. Although are inexact methods, the convergence rate of proximal-TMM is improved by factor 2 in the sense that proximal TMM has $\mathcal{O}\left(1-\sqrt{q}\right)^{2k}$ rate while the two algorithms have $\mathcal{O}\left(1-\sqrt{q}\right)^{k}$ rates, where $q$ is the inverse of the condition number.
>
>
> 5.	We are happy to hear that the reviewer found the overall writing to be good. Our arrow notation is indeed unusual in the context of optimization, but it is common in Euclidean geometry. Our figures may be unusual at first sight, but the illustrated geometric structure does reflect our reasoning. For example, in our analysis of Section 3.2, we use the illustration to make very clear which vectors are parallel and constant multiples of each other; in particular that $\overrightarrow{x_{k-1}^+ x_{k}}$ and $\overrightarrow{x_kz_k}$ are scalar multiples of each other would be much more difficult and timeconsuming without the geometric considerations.
>
>
> [1] Y. Nesterov, A. Gasnikov, S. Guminov, and P. Dvurechensky. Primal–dual accelerated gradient methods with small-dimensional relaxation oracle. Optimization Methods and Software, pages 1–38, 2020.
>
> [3] D. Kim and J. A. Fessler. Optimizing the efficiency of first-order methods for decreasing the gradient of smooth convex functions. Journal of Optimization Theory and Applications, 188(1):192–219, 2021.
>
> [8] A. d’Aspremont, D. Scieur, and A. Taylor. Acceleration methods. arXiv preprint arXiv:2101.09545, 2021
>
> [9] M. Barré, A. Taylor, and F. Bach. Principled analyses and design of first-order methods with inexact proximal operators. arXiv preprint arXiv:2006.06041, 2020.

---

### Official Review · Reviewer_YeUF · 2021-07-20

**Rating:** 7
**Confidence:** 3

**Summary:**

This paper studies the geometric structure in composite minimization by making gradients small. More specifically, the author identifies parallel and collinear geometric structures. The novelty of this paper is the analysis of parallel structure and its extension to new variants of accelerated methods. It is indeed important to study the geometric structures for convergence rates of gradients for convex problems and the Lyapunov analysis is a standard way to analyze the convergence rates.

**Limitations And Societal Impact:**

Yes.

**Main Review:**

This paper provides a comprehensive and elegant analysis in terms of providing geometric insights for accelerated methods. Basically, I think this paper is a wonderful work. The only two comments I have is : 1. It would be nice to have some theoretical insights for geometric structures under non-convex settings as citation[2] did. 2. It is encouraged to use more real dataset instead of generated ones to support the theoretical results in this paper.


**Time Spent Reviewing:**

6

---

> ### Author Response · Authors · 2021-08-09
> **Response to Reviewer YeUF**
>
> We are happy to hear that the reviewer found our work interesting.
>
> Thank you for the comments. We agree with the suggestion that using real, rather than synthetic, data would strengthen the experiments. Also, finding a geometric understanding of acceleration in non-convex setups is indeed an interesting direction, and we plan to pursue it in our future work.

---

### Decision · Program_Chairs · 2021-09-27

**Decision:**

Accept (Poster)

**Comment:**

The paper provides new geometric insights into acceleration under the Euclidean squared norm based on parallelism and collinearity of query and auxiliary points. The reviewers and area chair found the conceptual value of this contribution somewhat hard to evaluate, possibly because it only covers the Euclidean case and does not fully motivate the choice of Lyapunov function. However, the value of the contribution is reinforced by the fact that the authors use it to provide the best known bounds for making gradient small in the prox-grad setting and to derive a number of alternative algorithms for making gradient small. This improvement testifies to the novelty and usefulness of the techniques in the paper.